# DreamVLA: A Vision-Language-Action Model Dreamed with Comprehensive World Knowledge

**Wenyao Zhang**[124*]     **Hongsi Liu**[27*]     **Zekun Qi**[34*]     **Yunnan Wang**[12*]
**Xinqiang Yu**[4]     **Jiazhao Zhang**[45]     **Runpei Dong**[6]     **Jiawei He**[4]
**He Wang**[45]     **Zhizheng Zhang**[4]     **Li Yi**[3]     **Wenjun Zeng**[2]     **Xin Jin**[2‡]

[1]SJTU     [2]EIT     [3]THU     [4]Galbot     [5]PKU     [6]UIUC     [7]USTC

⌂ Project Page          Code          🤗 Hugging Face

## Abstract

Recent advances in vision-language-action (VLA) models have shown promise in integrating image generation with action prediction to improve generalization and reasoning in robot manipulation. However, existing methods are limited to challenging image-based forecasting, which suffers from redundant information and lacks comprehensive and critical world knowledge, including dynamic, spatial and semantic information. To address these limitations, we propose DreamVLA, a novel VLA framework that integrates comprehensive world knowledge forecasting to enable inverse dynamics modeling, thereby establishing a perception-prediction-action loop for manipulation tasks. Specifically, DreamVLA introduces a dynamic-region-guided world knowledge prediction, integrated with the spatial and semantic cues, which provide compact yet comprehensive representations for action planning. This design aligns with how humans interact with the world by first forming abstract multimodal reasoning chains before acting. To mitigate interference among the dynamic, spatial and semantic information during training, we adopt a block-wise structured attention mechanism that masks their mutual attention, preventing information leakage and keeping each representation clean and disentangled. Moreover, to model the conditional distribution over future actions, we employ a diffusion-based transformer that disentangles action representations from shared latent features. Extensive experiments on both real-world and simulation environments demonstrate that DreamVLA achieves **76.7%** success rate on real robot tasks and **4.44** average length on the CALVIN ABC-D benchmarks.

## 1 Introduction

The evolution of robot learning has demonstrated impressive progress [1–13] in training policies capable of performing diverse tasks across various environments [14–27]. One promising direction is Vision-Language-Action (VLA) models, which leverage the rich understanding capabilities of pre-trained Multimodal Large Language Models (MMLMs) [28–31] to directly map natural language instructions and visual observations to robot actions [17, 1, 14]. Although these approaches [32–34, 15, 1, 35–44] have achieved impressive results, their direct mapping from observations to actions lacks the closed-loop forecasting capability that humans typically possess when understanding and reasoning about future knowledge of environments.

To incorporate future knowledge prediction into VLA, most existing methods [45, 5, 46–57] leverage a copilot generation model to generate future frames/keypoints, then predict action sequences conditioned on goal images. Several methods [58–63] integrate pixel-level image forecasting with the

---

*Equal contribution. ‡Corresponding author.

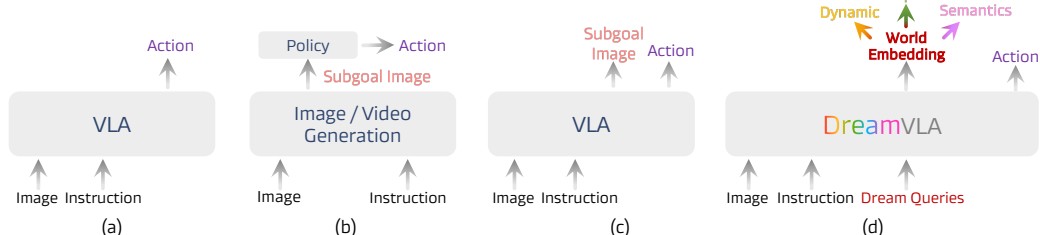

Figure 1: (a) Vanilla VLA directly maps visual observations and language instructions to actions. (b) Models leveraging separate image/video generation or copilot models to generate future frames or trajectories, subsequently guiding an action head. (c) VLA variants explicitly predict a subgoal image as an intermediate visual reasoning step prior to action generation. (d) Our proposed **DreamVLA**, which explicitly predicts dynamic regions, depth map, semantics (DINOv2 and SAM) knowledge, significantly enhances the model's action reasoning and generalization.

action prediction in a single framework, which exploits the synergy of prediction and planning and regards the prediction as an intermediate reasoning step [60] akin to those used in large language models (LLMs) [64]. Despite early success in incorporating dense visual forecasting, these methods naturally exhibit limitations: (1) *Redundant pixel information*: There exists significant overlap between forecasted images and current observations, making the prediction less efficient and effective. (2) *Lack of spatial information*: Absence of explicit 3D knowledge of environments [65–68, 24]. (3) *Lack of high-level knowledge forecasting*: Missing high-level understanding of future states, *e.g.*, semantics information. Therefore, we argue that existing methods (Figure 1 (a-c)) are insufficient to forecast future states for a more comprehensive prediction-action loop in the context of world-level future knowledge.

To address these issues, we propose DreamVLA, a novel framework that incorporates comprehensive world knowledge forecasting into the vision-language-action models, thereby establishing a perception-prediction-action loop for the manipulation task. As shown in Figure 1 (d), instead of directly generating entire future frames, our proposed method introduces *world embedding* to predict comprehensive world knowledge, which is highly relevant to robot execution, such as dynamic area, depth, and high-level semantic features. This approach aligns with the way humans interact with the world, emphasizing relevant changes and world knowledge. By dreaming/forecasting these targeted aspects of the environment, we aim to provide the model with concise and relevant intermediate representations that facilitate more effective action planning.

To obtain comprehensive world knowledge, our approach incorporates three key features: (1) *Dynamic region-based forecasting*. We leverage an off-the-shelf optical flow prediction model [69, 70] to identify dynamic regions within the scene, enabling the model to concentrate on areas of motion that are critical for task execution instead of redundant frame reconstruction. (2) *Depth-aware forecasting*. We employ depth estimation techniques [65] to generate per-frame depth maps, providing valuable spatial context that aids in understanding the three-dimensional structure of the environment. (3) *High-level foundation features*. We incorporate semantic features aligned with visual foundation models such as DINOv2 [71] and SAM [72]. In this way, DreamVLA offers a more comprehensive and effective pathway for the model to plan and execute. Furthermore, we adopt a block-wise structured attention mechanism that masks their mutual attention, preventing information leakage and keeping each representation clean and disentangled. Since the world and action embeddings occupy the same latent space and share similar statistics, a naive MLP head cannot disentangle modality-specific information or exploit their cross-modal correlations. We employ a diffusion-based transformer that disentangles action representations from shared latent features to reason actions.

Through extensive experiments on public benchmarks, we find that incorporating world knowledge prediction leads to significant performance improvements. Our method achieves state-of-the-art performance on the CALVIN benchmark (**4.44** average length), and we analyze the influence of the ingredients of our world knowledge and find that they have improvements in different aspects. Specifically, comprehensive ablation shows that predicting dynamic regions alone delivers the greatest gains, while depth and semantic cues offer smaller, roughly equal benefits. Worse, when depth or semantic prediction is used in isolation, it not only fails to help but can actually degrade performance. Extensive experiments on both simulation and real-world demonstrate the effectiveness of our method.

The key contributions of our work are summarized as follows:

- We recast the vision–language–action model as a perception–prediction–action model and make the model explicitly predict a compact set of dynamic, spatial and high-level semantic information, supplying concise yet comprehensive look-ahead cues for planning.

- We introduce a block-wise structured-attention mechanism, coupled with a diffusion-transformer decoder, to suppress representation noise from cross-type knowledge leakage and thus enable coherent multi-step action reasoning.

- DreamVLA sets a new state of the art on the CALVIN ABC-D benchmark (4.44 average task length), outperforming prior methods by up to 3.5% on the simulation platform, and boosts real-world success to 76.7%. Ablation studies confirm each component's contribution.

## 2 Related Works

### 2.1 Vision–Language–Action Models

The earliest VLA [18, 73, 2, 74–76] lay the foundation by combining pretrained vision-language representations with task-conditioned policies for manipulation and control. Inspired by the recent advances of Large Language Models [77–80] and multimodal large language models [30, 28, 81, 67, 82] and the emergence of large-scale robot datasets [14, 83–85], VLA has become a trend in robot learning. RT series [2, 86, 87] is the pioneer attempt to fine-tune the MLLM on robot demonstration datasets, resulting in strong accuracy and generalization. Building on this foundation, many advanced techniques [32, 34, 15, 1, 35, 36, 75, 37–39, 88–90, 40, 91] are developed to boost the performance. Meanwhile, considering the advantage of the diffusion model in modeling multi-peak, some researchers [92–96] employ different architectures to sample action from noise conditioned on observation, task instruction, and robot prior knowledge. Given on this manner which directly maps observation and instruction to action lacks reasoning steps like LLM [64], most existing methods [45, 5, 46–51] leverage a copilot image/video generation model to generate future frames then predict action sequences conditioned on goal images. However, the above methods still need an extra generation model, which introduces inference time and computation load. Therefore, several methods [58–63] integrate pixel-level forecasting with the action prediction in a single framework, which exploits the synergy of prediction and planning. Despite success, these methods naturally exhibit limitations in redundant reconstruction [97], and lack spatial and semantic information.

### 2.2 Knowledge Forecasting for Robotics

Learning future world knowledge for robot training has increasingly become popular to enable policies for achieving an action-forecasting loop. Early attempts [51, 21, 16, 45, 53, 52, 98] to implement this based on off-the-shelf video generation models [99, 55] and feed the goal images or states into policy model to conduct inverse dynamics. This two-stage training strategy is easy to implement but is limited by the performance and latency of video generation models. More advanced solutions couple forecasting with control by requiring the policy to produce, in addition to actions, explicit predictions. Concretely, these works ask the policy to output (i) high-level subtask/option sequences or language plans that decompose long-horizon goals [100–102], (ii)latent future embeddings/latent actions that compactly encode forthcoming motor intentions [90], (iii)whole sub-goal images or short visual rollouts that anticipate how the scene should evolve [58, 60], and (iv) object-centric signals (e.g., bounding boxes) that capture manipulation-relevant dynamics [85, 89]. This line of work demonstrates better performance and generalization. However, the future states are limited to redundant visual information [65, 66, 103, 71, 104, 68] or monotonous states [23, 50]. In contrast to previous work, DreamVLA proposes to predict future knowledge in an efficient (dynamic region) and effective (comprehensive knowledge) way, demonstrating strong performance and generalization.

## 3 Methodology

### 3.1 Problem Definition and Notation

We aim to improve robot execution by leveraging rich world knowledge as a guiding principle. In this context, we formulate vision–language–action reasoning as an *inverse dynamics* problem [105, 58, 51],

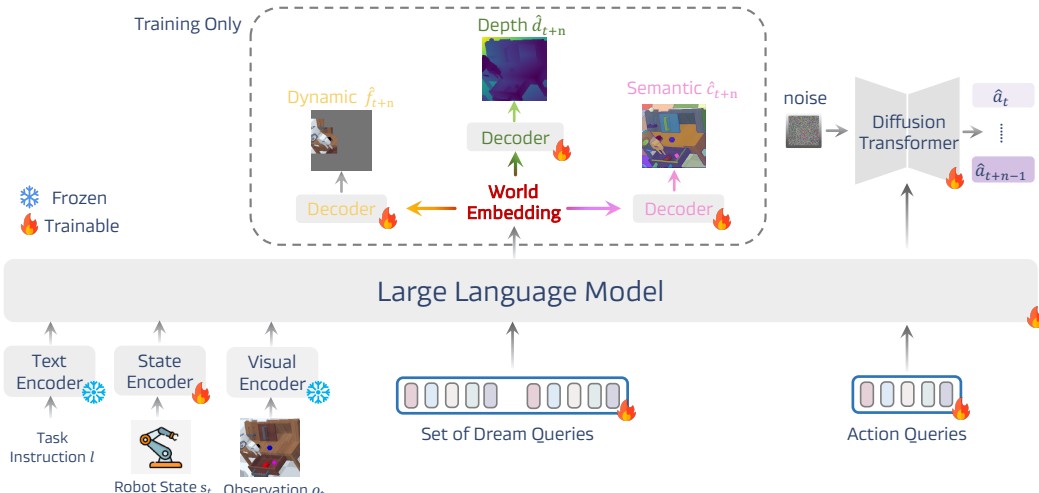

Figure 2: **Framework Overview**. Given the current robot state $s_t$, observation $o_t$, and language instruction, DreamVLA encodes multimodal inputs via frozen text, visual encoders and a tunable state encoder. These tokens, together with a learnable set of `<dream>` queries, are processed by a large language model to produce *world embedding*. Three lightweight decoders then project each corresponding element of this embedding into the dynamics region $\hat{f}_{t+n}$, monocular depth $\hat{d}_{t+n}$ and high-level semantics $\hat{c}_{t+n}$. A separate `<action>` query draws a latent action embedding, which conditions a diffusion transformer that refines Gaussian noise into an $n$-step action sequence $\hat{a}_{t:t+n-1}$. The dashed box highlights prediction heads that are used only during training; inference skips these heads and operates directly on the world embedding.

which regards the future world knowledge prediction as the intermediate reasoning for robot control, fully unleashing the synergy of prediction and execution. At each time step $t$, the robot receives three heterogeneous signals: a natural language instruction $l$, a raw visual frame $o_t$, and its proprioceptive state $s_t$. To inject look-ahead reasoning, we define a set of special tokens called `<dream>` queries [81], and concatenate all inputs into a sequence. A unified model $\mathcal{M}$ maps these inputs into a compact latent representation, which we call the *world embedding*:

$$\mathbf{w}_{t+n} = \mathcal{M}\left(l, o_t, s_t | \texttt{<dream>}\right). \tag{1}$$

Next, the world embedding predicts the comprehensive world knowledge that combines motion cues, spatial details and high-level semantics. Specifically, a set of predictor $\mathcal{P}$ extrapolates $n$ steps ahead,

$$\hat{p}_{t+n} = \mathcal{P}\left(\mathbf{w}_{t+n}\right) = \left[\hat{f}_{t+n}, \hat{d}_{t+n}, \hat{c}_{t+n}\right], \tag{2}$$

where $\hat{f}_{t+n}$ marks dynamic regions, $\hat{d}_{t+n}$ encodes monocular depth, and $\hat{c}_{t+n}$ optionally stores high-level semantic feature (e.g. DINOv2 [71], SAM [72]).

Given *world embedding* $\mathbf{w}_{t+n}$, the `<action>` query is assigned to the latent action embedding by the unified model $\mathcal{M}$ to aggregate the correlated action information. A denoising-diffusion transformer $\mathcal{D}$ formulates an $n$-step action based on the latent feature:

$$\hat{a}_{t:t+n-1} = \mathcal{D}(\mathcal{M}\left(l, o_t, s_t, \texttt{<dream>}|\texttt{<action>}\right)), \tag{3}$$

thus completing a perception–prediction–action loop that is identical during training and inference. The remainder of this chapter details the system components—encoders, world-knowledge predictor, and diffusion-based action generator—that instantiate the above formulation.

## 3.2 Model Architecture

As illustrated in Figure 2, our DreamVLA framework comprises three core modules operating within a unified transformer architecture. Firstly, heterogeneous inputs—including natural language $l$, visual observations $o_t$, and proprioceptive states $s_t$—are individually processed by modality-specific encoders. We encode language instructions using CLIP [103] text embeddings, visual frames through a Masked Autoencoder [106] to obtain spatiotemporal patch representations, and proprioceptive signals via several convolutional and fully-connected layers. Following encoding, a set of learnable

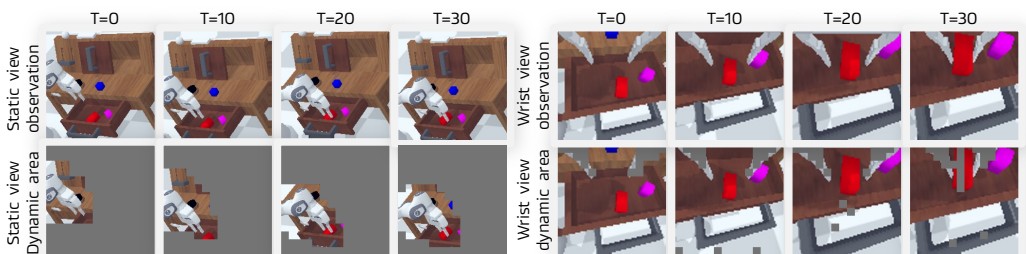

Figure 3: **Visualization of dynamic regions over time.** We show the static camera (left) and wrist-mounted camera (right) observations alongside the corresponding dynamic masks generated by our method at multiple time steps. The masks highlight dynamic regions by leveraging optical flow trajectories extracted via CoTracker [70, 69]. Compared to the original observations, our method effectively suppresses irrelevant background and focuses on interaction-relevant areas (e.g., moving objects and end-effector), enabling more structured and efficient action reasoning.

queries designated as `<dream>` and `<action>` are appended to these multimodal embeddings, where `<dream>` contains three subqueries (dynamic, depth and semantics), which could be used for the prediction of specific knowledge. Subsequently, we leverage a large language model based on GPT-2 [107] to integrate and attend across modalities and queries using carefully structured causal and non-causal attention mechanisms (Figure 4). This effectively fuses low-level perceptual signals into compact, semantically coherent representations of the world state.

Finally, specialized light-weight output heads comprising by shallow convolutional layers decode *world embedding* into explicit predictions: reconstruct anticipated dynamic region, monocular depth, and semantic features. During inference, DreamVLA skips the decoder entirely, saving substantial computation. Instead, the model outputs an *world embedding* that encapsulates predictions of future dynamics, depth, and semantics without pixel-level reconstruction, thereby retaining the accuracy gains from future-state reasoning while maintaining low latency. In parallel, we employ a denoising diffusion transformer [92] to decode latent action embedding into executable robot action sequences. Collectively, these components enable DreamVLA to perform robust, predictive vision–language–action reasoning in an end-to-end manner.

### 3.3 Comprehensive World Knowledge Prediction

Predicting *what will matter next* is more valuable than merely reproducing the raw future frame. DreamVLA explicitly forecasts future world knowledge that is most relevant for manipulation, including (i) motion–centric **dynamic region**, (ii) 3D **depth geometry**, and (iii) high-level **semantics**. These complementary signals provide a compact, structured surrogate for raw pixels and supply the policy with look-ahead context for inverse dynamics planning.

**Motion-centric dynamic-region reconstruction.** Predicting dynamic regions tells the robot *what parts of the scene are about to move*, allowing the model to capture the statistical link between the current scene, the language instruction, and the actions needed to realize the predicted motion. As shown in Figure 3, DreamVLA neither predicts dense optical flow nor synthesizes an entire future frame. Instead, we first apply CoTracker [69, 70] to extract dynamic regions, namely pixels that move with the robot end-effector or other movable objects, and then train DreamVLA to reconstruct only these regions. Furthermore, generating reconstruction targets with an asymmetrical tokenizer can further enhance performance [106]. From the perspective of discrete variational autoencoder (dVAE) [108–111], the overall optimization is to maximize the *evidence lower bound* (ELBO) [112–114, 68] of the log-likelihood $P(x_i|\tilde{x}_i)$. Let $x$ denote the original image, $\tilde{x}$ the masked motion region, and $z$ the reconstruction target. The generative modeling can be described as:

$$\sum_{(z_i,\tilde{z}_i)\in\mathcal{D}} \log P(x_i|\tilde{x}_i) \geq \sum_{(x_i,\tilde{x}_i)\in\mathcal{D}} \Big( \mathbb{E}_{z_i\sim Q_\phi(\mathbf{z}|x_i)}\big[\log P_\psi(x_i|z_i)\big] - D_{\mathrm{KL}}\big[z, P_\theta(\mathbf{z}|\hat{z}_i)\big]\Big), \quad (4)$$

where $P_\psi(x|z)$ is the tokenizer decoder to recover origin data, $\hat{z}_i = Q_\phi(\mathbf{z}|\tilde{x}_i)$ denotes the masked motion region tokens from masked data and $P_\theta(z|\hat{z}_i)$ reconstructs masked tokens in an autoencoding

fashion. Here, the $P_\theta(z|\hat{z}_i)$ is zero, and the dynamic region prediction loss can be formulated as:

$$\mathcal{L}_{\text{dyn}} = \frac{1}{|\mathcal{D}|} \sum_{x_i \in \mathcal{D}} \mathbb{E}_{z \sim Q_\phi(z|x_i)} \Big[ -\log P_\psi\big((x_i)_{\mathcal{M}} \mid z\big) \Big]. \tag{5}$$

**Depth prediction.** Predicting how the depth field will evolve tells the robot *where it should move next*, steering it toward free space and away from impending obstacles. If depth sensors are available, we supervise the DreamVLA with ground-truth maps; on low-cost platforms without depth sensing, we instead hallucinate future geometry from a single RGB stream. To do so, we treat Depth-Anything [65, 66] predictions as a self-supervised teacher and train a dedicated *depth query* to regress the aligned future map $\hat{d}_{t+n}$. The objective is a scale-normalized mean-squared error,

$$\mathcal{L}_{\text{depth}} = \frac{1}{HW} \sum_{i,j} \big( \hat{d}_{t+n}^{(i,j)} - \alpha\, d_{t+n}^{(i,j)} \big)^2, \tag{6}$$

$$\alpha = \frac{\sum_{i,j} \hat{d}_{t+n}^{(i,j)} d_{t+n}^{(i,j)}}{\sum_{i,j} d_{t+n}^{(i,j)\,2}}, \tag{7}$$

where $\alpha$ removes the global scale ambiguity that monocular methods cannot resolve. In practice, this simple loss is sufficient: the teacher provides metrically plausible depth, and the scale-normalization term encourages the model to preserve ordinal depth relationships, a property that is crucial for grasp synthesis and collision checking, while ignoring any arbitrary global scale shift.

**Contrastive semantic forecasting.** Predicting future semantics teaches the robot *which objects or regions will matter for the task*, providing a high-level context (for example, object identity and affordances) that guides the selection of goals and grasp choice. To learn these semantics, DreamVLA predicts future DINOv2 [71] and SAM [72] feature $\hat{c}_{t+n}$ using an InfoNCE loss [115, 68]: the ground-truth feature is the positive sample, whereas spatially shifted features act as negatives. This encourages discriminative anticipation that the model must pick the correct object semantics among plausible but wrong futures:

$$\mathcal{L}_{\text{sem}} = -\log \frac{\exp\big(\hat{c}_{t+n}^\top c_{t+n}/\tau\big)}{\sum_k \exp\big(\hat{c}_{t+n}^\top c_k/\tau\big)}, \tag{8}$$

where $k$ represents the number of tokens in spatial, and $\tau$ denotes the temperature.

**Structured attention for cross-type knowledge disentanglement.** To preserve clear cross-type knowledge boundaries, `<dream>` is decomposed into three sub-queries (dynamic, depth and semantics). If these sub-queries could freely attend to one another, high-frequency flow details would contaminate depth reasoning, and semantic cues might bleed into motion features, producing noisy mixed representations. We therefore mask their mutual attention: each sub-query attends only to the shared visual, language, and state tokens, while direct links among the three are disabled, keeping their latent features disentangled and free of cross-talk. As shown in Figure 4, both `<dream>` and `<action>` queries also employ causal attention restricted to past context, which preserves temporal causality. This organized pattern mirrors the specialist routing used in Mixture-of-Experts (MoE) networks [116]. By avoiding cross-modal leakage,

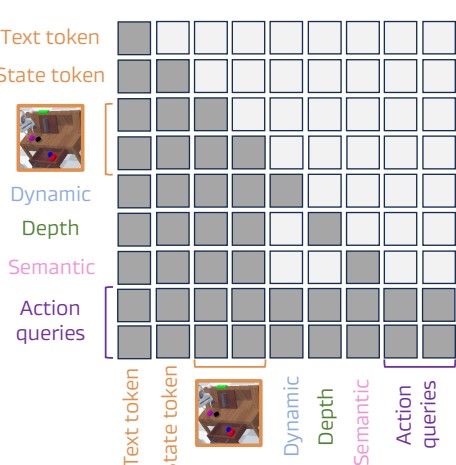

Figure 4: Block-wise structured attention.

the structured attention supplies clean future world knowledge for action prediction, improves robustness, and maintains temporal consistency.

## 3.4 Inverse Dynamics via Denoising Diffusion Transformer

Given two ordered observations $o_t$ and $o_{t+1}$, classical inverse dynamics infers the intermediate action $\hat{a}_t$. We extend this formulation by predicting a full action sequence $\hat{a}_{t:t+n-1}$ conditioned

Table 1: **CALVIN ABC-D results.** We present the average success computed over 1000 rollouts for each task and the average number of completed tasks to solve 5 instructions consecutively (Avg. Len.). DreamVLA shows significant superiority over baselines. The best results are **bolded**.

| Method | Task completed in a row | | | | | Avg. Len. ↑ |
|---|---|---|---|---|---|---|
| | 1 | 2 | 3 | 4 | 5 | |
| Roboflamingo [32] | 82.4 | 61.9 | 46.6 | 33.1 | 23.5 | 2.47 |
| Susie [120] | 87.0 | 69.0 | 49.0 | 38.0 | 26.0 | 2.69 |
| GR-1 [16] | 85.4 | 71.2 | 59.6 | 49.7 | 40.1 | 3.06 |
| 3D Diffusor Actor [95] | 92.2 | 78.7 | 63.9 | 51.2 | 41.2 | 3.27 |
| OpenVLA [1] | 91.3 | 77.8 | 62.0 | 52.1 | 43.5 | 3.27 |
| RoboDual [121] | 94.4 | 82.7 | 72.1 | 62.4 | 54.4 | 3.66 |
| UNIVLA [122] | 95.5 | 85.8 | 75.4 | 66.9 | 56.5 | 3.80 |
| $\pi_0$ [34] | 93.8 | 85.0 | 76.7 | 68.1 | 59.9 | 3.84 |
| CLOVER [123] | 96.0 | 83.5 | 70.8 | 57.5 | 45.4 | 3.53 |
| UP-VLA [59] | 92.8 | 86.5 | 81.5 | 76.9 | 69.9 | 4.08 |
| Robovlm [39] | 98.0 | 93.6 | 85.4 | 77.8 | 70.4 | 4.25 |
| Seer [58] | 96.3 | 91.6 | 86.1 | 80.3 | 74.0 | 4.28 |
| VPP [51] | 95.7 | 91.2 | 86.3 | 81.0 | 75.0 | 4.29 |
| DreamVLA | **98.2** | **94.6** | **89.5** | **83.4** | **78.1** | **4.44** |

on the current observation $o_t$ and future latent world embeddings $\mathbf{w}_{t+n}$. Specifically, DreamVLA first aggregates this latent embedding, already enriched with predicted future dynamics, depth, and semantics, into a compact action embedding via a dedicated action query and the model's causal attention. Since the world and action embeddings occupy the same latent space and share similar statistics, a naive MLP head cannot disentangle modality-specific information or exploit their cross-modal correlations. We therefore employ a denoising diffusion transformer (DiT) [92, 117] as the action head. Conditioned on the action embedding, DiT employs iterative self-attention and denoising to fuse perceptual forecasts with control priors and to transform Gaussian noise into an $n$-step trajectory $a_{t:t+n-1}$, yielding coherent, diverse, and physically grounded action sequences. The loss of action prediction can be formulated as:

$$\mathcal{L}_{\text{DiT}} = \mathbb{E}_{\tau,\varepsilon}\big\|\varepsilon - \varepsilon_\theta\big(\sqrt{\bar{\alpha}_\tau}\, a_{t:t+n-1} + \sqrt{1-\bar{\alpha}_\tau}\, \varepsilon, \tau, \mathbf{c}\big)\big\|_2^2, \tag{9}$$

where $\varepsilon_\theta$ is the DiT denoiser, $\varepsilon \sim \mathcal{N}(0, I)$, $\bar{\alpha}_\tau$ follows a cosine noise schedule and $\mathbf{c}$ is the latent action embedding obtained from a large language model. Inference is performed by drawing a Gaussian sample and running the learned reverse diffusion, yielding diverse yet physically plausible trajectories that close the perception–prediction–action loop.

## 4 Experiments

### 4.1 Implementation Details

All models are implemented in PyTorch and trained on NVIDIA 8 A800 GPUs. We use an AdamW [118] optimizer with initial learning rate $10^{-3}$, weight decay $1e-4$, and a cosine learning-rate schedule with 5% linear warm-up. Batch size is set to 64, we set the query length of each modality 9 and diffusion steps in DiT to 10. We weight the dynamic region, depth and segmentation prediction losses as $\lambda_{\text{dyn}}=0.1$, $\lambda_{\text{depth}}=0.001$, $\lambda_{\text{sem}}=0.1$, and the action loss as $\lambda_{\text{DiT}}=1$, respectively. We first pre-train DreamVLA on the language-free split of the CALVIN [119] and on the full DROID dataset [84]. For the LIBERO benchmark, we first pretrain DreamVLA on LIBERO-90 and then finetune on each track. The model predicts entire frames instead of comprehensive knowledge, keeping storage and computation requirements manageable. We then fine-tune DreamVLA on each target dataset using the comprehensive world knowledge forecasting objective. All models are trained for 20 epochs, and we select the checkpoint with the highest validation success rate (SR) for final evaluation.

### 4.2 Simulation Benchmark Experiments

**Simulation setup.** We evaluate DreamVLA on CALVIN [119] and LIBERO [124] benchmark. CALVIN is a simulated benchmark designed for learning long-horizon, language-conditioned robot manipulation policies. It comprises four distinct manipulation environments and over six hours

Table 2: **The extended LIBERO experiments.** DreamVLA achieves the best or competitive performance across all tracks compared to previous approaches. The best results are **bolded**.

| Methods | Scores (%) | | | | Average |
|---|---|---|---|---|---|
| | Spatial | Object | Goal | Long | |
| Diffusion Policy [92] | 78.3 | 92.5 | 68.3 | 50.5 | 72.4 |
| Octo [15] | 78.9 | 85.7 | 84.6 | 51.1 | 75.1 |
| OpenVLA [1] | 84.7 | 88.4 | 79.2 | 53.7 | 76.5 |
| SpatialVLA [38] | 88.2 | 89.9 | 78.6 | 55.5 | 78.1 |
| CoT-VLA [60] | 87.5 | 91.6 | 87.6 | 69.0 | 81.1 |
| DreamVLA | **97.5** | **94.0** | **89.5** | **89.5** | **92.6** |

of teleoperated play data per environment, captured from multiple sensors including static and gripper-mounted RGB-D cameras, tactile images, and proprioceptive readings. We report the success rate of every track and the average length of 5 tasks. Additionally, evaluations are also conducted on LIBERO [124], a simulated benchmark spanning four suites (LIBERO-Spatial/-Object/-Goal/-Long). Each suite contains 10 tasks supported by 50 human-teleoperated demonstrations, targeting spatial reasoning, object-centric manipulation, and goal completion.

**Results.** As shown in Table 1, DreamVLA achieves the highest performance on ABC-D tasks, Our method surpasses Roboflamingo [32], 3D Diffusor Actor [95], OpenVLA [1], RoboDual [121], UNIVLA [122], Robovlm [39] and GR1 [16], which directly projects the RGB/depth image to action signals as shown in Figure 1(a) in the manuscripts. Compared to methods that use a copilot model to generate sub-goal images as input, like Susie [120] and CLOVER [123] as shown in Figure 1(b) in manuscripts, our model significantly achieves more accurate control. DreamVLA outperforms approaches like UP-VLA [59], Seer [58], and VPP [51] as shown in Figure 1(c), which merge whole sub-goal image foresight into one VLA to take benefits from a more integrated design and joint optimization. indicating that our method has better multi-task learning and generalization capabilities in simulation tasks. For the LIBERO benchmark [124], DreamVLA exhibits better or comparable ability across all tracks compared to previous approaches by future world knowledge prediction as shown in Table 2.

## 4.3 Real World Experiments

To evaluate the effectiveness of our method in the real-world, we use the Franka Panda arm to conduct real-world experiments on gripper grasping. In our setups, two RealSense D415 cameras capture RGB images. One is in a third-person view, and the other is at the end of the robotic arm, as shown in Figure 5. We collect four categories of objects for two tasks: pick and place. Additionally, we conduct experiments on drawer opening and closing tasks, as shown in the supplementary. Follow [58], we pretrain DreamVLA on the DROID [84] contains large-scale trajectories of Franka robots in varied scenes. For fair comparison, we fine-tune Diffusion Policy [92], Octo-Base [15], OpenVLA [1] and DreamVLA on collected demonstration datasets containing 100 trajectories for each task.

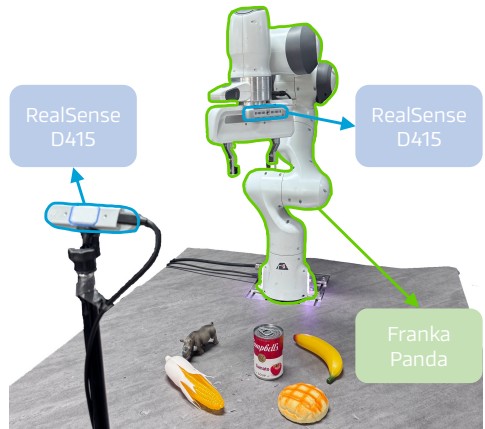

Figure 5: Real-world experiment setup.

In the experimental setup, each trial permits a maximum of 20 consecutive attempts. For the grasping experiments, objects are randomly positioned on the table surface. A trial is deemed successful if the robotic arm successfully grasps the target object within the predefined attempt limit. In the placement experiments, the robot is required to transfer the grasped object into a designated basket. Success is recorded only if both the grasping and placement operations are completed within the allowed attempts. For the drawer manipulation tasks, the drawer is placed randomly in front of the robotic arm. The experiment is considered successful if the drawer displacement exceeds 10 centimeters, indicating effective interaction. The results, presented in Table 3, demonstrate that our method performs better than other methods. More real-world experiment visualizations are shown in the supplementary section.

Table 3: **Real-world evaluation** with the Franka Robot across three tasks.

| Method | Pick | | | Place | | | Drawer | | | Task (All) |
|---|---|---|---|---|---|---|---|---|---|---|
| | Bottle | Doll | Avg. | Banana | Chili | Avg. | Open | Close | Avg. | Avg. |
| Diffusion Policy [92] | 50.0 | 70.0 | 60.0 | 65.0 | 45.0 | 55.0 | 15.0 | 60.0 | 37.5 | 50.8 |
| Octo-Base [15] | 50.0 | 60.0 | 55.0 | 40.0 | 50.0 | 45.0 | 20.0 | 50.0 | 35.0 | 45.0 |
| OpenVLA [1] | 50.0 | 40.0 | 45.0 | 20.0 | 30.0 | 25.0 | 40.0 | 30.0 | 35.0 | 35.0 |
| DreamVLA | **85.0** | **80.0** | **82.5** | **80.0** | **80.0** | **80.0** | **70.0** | **65.0** | **67.5** | **76.7** |

Table 4: **Performance comparison** between predicting the optical flow and the dynamic region. Notably, the * denotes that this result is from [58].

| Method | Task completed in a row | | | | | |
|---|---|---|---|---|---|---|
| | 1 | 2 | 3 | 4 | 5 | Avg. Len. ↑ |
| Vanilla VLA* | 93.0 | 82.4 | 72.3 | 62.6 | 53.3 | 3.64 |
| + dynamic region | 97.6 | 92.6 | 87.5 | 80.4 | 73.7 | 4.32 |
| + depth | **98.3** | 94.3 | 88.5 | 82.0 | 77.2 | 4.40 |
| + semantics | 98.2 | **94.6** | **89.5** | **83.4** | **78.1** | **4.44** |

## 4.4 Ablation Study

In this section, we design the experiments to investigate the following questions.
**Q1: What is the contribution of each modal characteristic?**

The core motivation of DreamVLA is to enable the model to predict comprehensive visual knowledge of the future to enhance action reasoning. However, not all types of knowledge contribute equally to subsequent execution. We consider four types of predictive knowledge: dynamic region, depth, and semantic segmentation features derived from SAM and DINO. As shown in Figure 6, we first train the model with each knowledge forecasting independently. The green dashed line denotes the performance of the Vanilla VLA baseline, which uses no knowledge prediction. Among all, predicting dynamic regions proves to be the most beneficial, because these masks explicitly flag the pixels that are about to change and therefore align almost perfectly with the policy's action semantics. By contrast, supervising the network with depth map, DINO or SAM features alone not only fails to help but often degrades performance. We analyze that this gap stems from how closely each auxiliary target matches the downstream objective: dynamic-region labels supply gradients that reinforce the action head, whereas depth regression and high-dimensional feature matching (DINO/SAM) inject large, noisy losses that dominate optimization. With the limited model attention budget, these competing gradients dilute the task-relevant features and push the backbone toward suboptimal optima, producing the observed drop below the dashed baseline.

Next, we train the model with all five knowledge heads simultaneously (All) and perform an ablation study (All-X), where we remove one knowledge signal at a time to evaluate its contribution. Removing F leads to the most significant performance drop, confirming its essential role. Interestingly, removing DINO results in similar or even better performance, suggesting that not all semantic signals are equally helpful or stable in predicting outcomes, so we only use semantic features from SAM in the subsequent ablations. Table 4 reveals a clear and decreasing return pattern in all ablations.

**Q2: Auxiliary Tasks vs. Future Knowledge Prediction: which drives improvement?**

Table 5 contrasts two training regimes: predicting complete world knowledge and performing auxiliary reconstructions, showing that the former is decisively superior. In our ablation, every prediction strategy is individually replaced by its reconstruction counterpart, yet each substitution consistently lowers performance: VLA trained only to redraw the current RGB, depth, semantics, or DINOv2 features can handle the first few actions but soon loses coherence, whereas a network trained to forecast the next dynamic region, depth map, and semantics preserves accuracy throughout the trajectory and carries tasks much farther before failure. The reason is that prediction provides a richer, action-oriented signal, directing learning toward the pixels that will drive the upcoming decision, while reconstruction merely revisits background detail that the control policy never actually needs.

**Q3: Why do we use the optical flow as the mask instead of directly forecasting it?**

To justify our choice of employing motion-centric dynamic regions over direct flow forecasting, we implement both variants under identical settings (Table 6). In the optical flow setup, the model must predict the full future flow field along with the subgoal image, which significantly increases

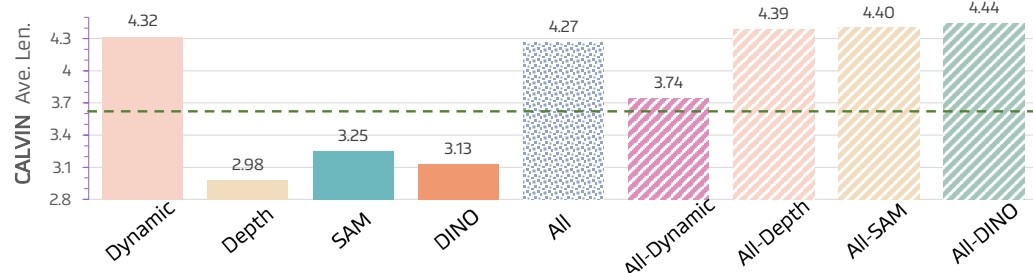

Figure 6: CALVIN ABC-D performance with respect to different combinations of knowledge prediction. All=all of five models, and All-X=taking X out of All.

Table 5: **Performance comparison** between co-training with auxiliary tasks and predicting the comprehensive world knowledge.

| Method | Task completed in a row | | | | | |
|---|---|---|---|---|---|---|
| | 1 | 2 | 3 | 4 | 5 | Avg. Len. |
| Auxiliary | 97.7 | 92.3 | 85.6 | 79.5 | 74.2 | 4.14 |
| Prediction | **98.2** | **94.6** | **89.5** | **83.4** | **78.1** | **4.44** |

Table 6: **Performance comparison** between predicting the optical flow and dynamic region.

| Method | Task completed in a row | | | | | |
|---|---|---|---|---|---|---|
| | 1 | 2 | 3 | 4 | 5 | Avg. Len. |
| Optical | 97.6 | 92.4 | 86.8 | 81.7 | 75.4 | 4.23 |
| Dynamic | **98.2** | **94.6** | **89.5** | **83.4** | **78.1** | **4.44** |

Table 7: **Performance comparison** between vanilla causal and our structured attention.

| Method | Task completed in a row | | | | | |
|---|---|---|---|---|---|---|
| | 1 | 2 | 3 | 4 | 5 | Avg. Len. |
| Causal | 94.2 | 86.5 | 78.4 | 71.3 | 62.7 | 3.75 |
| Structure | **98.2** | **94.6** | **89.5** | **83.4** | **78.1** | **4.44** |

Table 8: **Performance comparison** between shared and seprated queries.

| Method | Task completed in a row | | | | | |
|---|---|---|---|---|---|---|
| | 1 | 2 | 3 | 4 | 5 | Avg. Len. |
| Shared | 95.5 | 90.1 | 83.8 | 76.9 | 70.4 | 4.17 |
| Separated | **98.2** | **94.6** | **89.5** | **83.4** | **78.1** | **4.44** |

the training complexity. This extra burden manifests in markedly lower multi-step success rates. By contrast, our dynamic region approach merely employs the pretrained flow model to obtain a binary mask, focusing the model on "where" relevant motion occurs, bringing a significant improvement.

**Q4: The effectiveness of structured attention in DreamVLA.**

To demonstrate the effectiveness of our proposed structure attention mechanism in Figure 4, we swap it for a vanilla causal mask while keeping everything else fixed. In this setting, every <dream> query, including the one meant to capture semantics, can also read the flow and depth tokens produced in the same step; the extra cross-peek mixes unrelated signals, adds gradient noise, and quickly degrades long-horizon control. Our mask removes all query-to-query edges, so <action> query consults only past language, state and multimodal predictions, never their siblings. Table 7 shows the payoff: the causal variant brings a marginal improvement for Vanilla VLA, whereas the block-sparse version keeps success high throughout, confirming that blocking intra-step leakage is important.

**Q5: Can we use the shared query to predict the comprehensive world knowledge?**

Instead of assigning separate queries to dynamic region, depth, and semantics features, one might let a single set of shared queries predict all signals. To test this idea, we split each world-embedding vector into four equal sub-spaces, with each quarter intended to carry a different modality. Table 8 shows that the shared-query design hurts action performance: mixing modalities in the same query introduces cross-talk, so the diffusion head receives noisy features. In contrast, giving each modality its query keeps the representations disentangled and yields a clear performance gain.

## 5 Conclusion

We present DreamVLA, a novel Visual-Language-Action framework that enables inverse dynamics modeling through comprehensive world knowledge prediction, supporting the perception-prediction-action loop for manipulation tasks. DreamVLA leverages dynamic-region-guided knowledge forecasting, combining spatial and semantic cues to generate compact and informative representations for action planning. We introduce a block-wise structured-attention mechanism, coupled with a diffusion-transformer decoder, to suppress representation noise from cross-type knowledge leakage and thus enable coherent multi-step action reasoning. Extensive experiments in both real and simulated environments demonstrate the effectiveness of DreamVLA, achieving a 76.7% success rate on real-world robot tasks and outperforming prior methods on the CALVIN ABC-D benchmark.

## Acknowledgements

This work was supported by Grants of NSFC 62302246, ZJNSFC LQ23F010008, Ningbo 2023Z237 & 2024Z284 & 2024Z289 & 2023CX050011 & 2025Z038 & 2025Z091 and supported by High Performance Computing Center at Eastern Institute of Technology and Ningbo Institute of Digital Twin.

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

# Appendix

# A    Implementation Details

## A.1    DreamVLA Architecture

**Text Encoder.**    We use the CLIP ViT-B/32 text encoder [103] to process natural language task instructions. The encoder transforms each instruction into a fixed-length embedding that captures semantic intent. These embeddings are then projected into the shared latent space and used to condition the subsequent modules, enabling effective grounding of language into perception and action.

**Visual Encoder.**    We employ an MAE-pretrained ViT-B [106] as the vision encoder. At each timestep, images are captured from two views: eye-on-hand and eye-on-base. Each image is processed by the vision encoder to produce 196 latent vectors, which represent local patch information, along with a [CLS] token that encodes the global representation of the image. Directly inputting all 197 tokens into the transformer backbone would create a significant computational burden, particularly when processing long histories. Moreover, many image details are redundant for accomplishing manipulation tasks. To address this, we utilize the Perceiver Resampler [125] to condense the image representations and extract task-relevant features. The Perceiver Resampler employs learnable latent vectors with a shape of (num latents, dim), where num latents is significantly smaller than the number of image tokens. Through Perceiver Attention, these latent vectors condense the input image features, along with the [CLS] token, to form the final image tokens.

**Robot State.**    The robot state consists of the arm and gripper state. The arm state includes the end-effector position and its rotation in Euler angles, resulting in a six-dimensional representation. The gripper state is a binary value indicating whether the gripper is open or closed. We tokenize the robot state using an MLP. Specifically, the gripper state is first converted into a one-hot encoding. The one-hot encoding of the gripper state and the arm state are then each passed through separate linear layers. The outputs are concatenated and passed through a final linear layer to produce the state token.

**Learnable Queries.**    We introduce two sets of learnable query tokens, denoted as `<dream>` and `<action>`, to extract and integrate information from multimodal inputs for joint prediction.

The `<dream>` queries provide structured supervision through comprehensive knowledge prediction tasks and consist of 64 tokens in total, organized as 9 queries for each of the three modalities: dynamic motion, depth estimation and semantic features. These queries guide the model in reconstructing rich visual representations, enhancing the quality of the learned latent space.

The `<action>` query is dedicated to action sequence prediction. Their length is determined by the temporal prediction horizon, as defined in the action chunking strategy from [74].

**Large Language Model.**    We adopt GPT-2 Medium [107] as our language backbone. GPT-2 Medium is a 24-layer, 16-head Transformer decoder with a hidden size of 1,024 and a total of approximately 345 million parameters. It was pretrained on the WebText corpus ($\sim$8 million documents, 40 GB of text) using autoregressive language modeling to predict the next token with a byte-pair encoding vocabulary of 50,257 tokens.

**Output Heads.**    To decode the *world embedding* into comprehensive world knowledge, we incorporate multiple task-specific output heads that predict dynamic motion regions, depth maps, and high-level semantics, including DINOv2 [71] and SAM-style segmentation features [72].

Each prediction head is implemented using a lightweight Vision Transformer (ViT) decoder, which operates on two types of tokens produced by the multimodal backbone: the latent embeddings associated with a specific modality and a set of learnable mask tokens used for reconstruction.

To retain spatial correspondence, we inject fixed sine–cosine positional encodings into the token embeddings. These tokens are then processed through several Transformer encoder layers, followed by a modality-specific linear projection head that maps each patch token to its output space, such as per-pixel depth values or semantic logits—thereby reconstructing the expected visual signals of future observations. Concrete details of each module are shown in Table 9.

**Action Prediction with Diffusion Transformer**    To generate future actions conditioned on latent action embeddigns, we adopt a diffusion-based Transformer architecture, DiT-B [117], as our action decoder. DiT enables flexible modeling of complex action distributions by progressively denoising a sequence of latent action tokens through a series of Transformer layers, allowing the model to capture multimodal uncertainty in robot control.

Table 9: The parameters of the each module in DreamVLA.

|                     | Hidden size | Number of layers | Number of heads |
|---------------------|-------------|------------------|-----------------|
| image encoder       | 768         | 12               | 12              |
| perceiver resampler | 768         | 3                | 8               |
| LLM                 | 1024        | 24               | 16              |
| image decoder       | 1024        | 2                | 16              |
| depth decoder       | 1024        | 2                | 16              |
| semantic decoder    | 1024        | 2                | 16              |

We configure the DiT model with the base variant (DiT-B), using an action token embedding size equal to the hidden dimension of the fusion Transformer. The model predicts $K$ future actions, where each action is a 7-dimensional vector that encodes the displacement of the pose and gripper state of the end effector. In our experiments, we set $K = 2$, corresponding to a 3-frame prediction window (current + 2 future steps). The model does not utilize past action context during generation (i.e., past window size is 0), focusing solely on predictive synthesis.

During training, Gaussian noise is added to the future action trajectories, and the model learns to reverse this corruption process step by step. This module operates on top of the aggregated representation via `<action>` query, enabling temporally coherent and semantically grounded action generation. The concrete detail of DiT is shown in Table 10.

Table 10: Configuration of the DiT-B model used for action prediction.

| Parameter | Value |
|-----------|-------|
| Model type | DiT-B |
| Token size | 1024 |
| Action prediction window | 2 future steps (3-frame chunk) |
| Past context steps | 0 |
| Number of Transformer layers | 12 |
| Number of attention heads | 12 |
| Positional encoding | Learned (1D for time) |
| Diffusion timesteps (Train) | 8 |
| Diffusion timesteps (Inference) | 10 |
| Noise schedule | Linear |
| Loss function | Denoising Score Matching (L2 loss) |
| Precision | float32 |

## A.2   Feature Extraction

To facilitate dynamic region prediction, we adopt a motion-based heuristic to generate coarse binary masks that highlight regions of interest. Given a sequence of consecutive RGB frames of resolution $H \times W$, we uniformly sample one keypoint every 8 pixels in both spatial dimensions, resulting in $N = \lfloor H/8 \rfloor \times \lfloor W/8 \rfloor$ sampled locations per frame. For each sampled location, we compute inter-frame displacements $(\Delta x, \Delta y)$ by tracking its position across adjacent frames using CoTracker [69]. The magnitude of displacement is converted into a scalar speed value:

$$s_{ij} = \sqrt{(\Delta x_{ij})^2 + (\Delta y_{ij})^2},$$

where $(i, j)$ denotes the spatial coordinates of each sampled patch. We then apply a speed threshold $\tau$ (e.g., $\tau = 1$ pixel/frame) to obtain a binary motion mask. To account for small motions and ensure spatial connectivity, we perform a single-pixel morphological dilation, expanding each positive location to its eight-connected neighbors.

The resulting mask is flattened and reshaped into the form $(B, 1, L)$, where $L = \lfloor H/8 \rfloor \cdot \lfloor W/8 \rfloor$ and $B$ is the batch size. We apply this binary mask element-wise to both predicted patch embeddings $\{\hat{p}_i\}$ and their corresponding ground-truth embeddings $\{p_i\}$ during loss computation, encouraging accurate representation in dynamic regions.

For depth supervision, we use the ground-truth depth maps provided by datasets when available. In cases where depth annotations are not provided—such as in certain real-world robot datasets—we use monocular depth estimators, specifically Depth-Anything v2 [66], to generate pseudo-ground-truth depth labels.

In addition to depth and dynamic signals, we include high-level feature supervision. For DINOv2 [71], we extract features from the final transformer layer, capturing global semantic and structural representations. For SAM [72], we utilize the output of its image encoder as dense segmentation-aware features. These diverse modalities collectively provide comprehensive supervision signals to improve the quality and generalizability of our learned visual representations.

### A.3 Training Detail

The total loss can be formulated as:

$$\mathcal{L} = \lambda_{\text{dyn}}\mathcal{L}_{\text{dyn}} + \lambda_{\text{depth}}\mathcal{L}_{\text{depth}} + \lambda_{\text{sem}}\mathcal{L}_{\text{sem}} + \lambda_{\text{DiT}}\mathcal{L}_{\text{DiT}} \qquad (10)$$

where $\lambda_{\text{dyn}} = 0.1, \lambda_{\text{depth}} = 0.001, \lambda_{\text{sem}} = 0.1, \lambda_{\text{DiT}} = 1$.

We train DreamVLA on 8 NVIDIA A800 GPUs. The main bottleneck is the memory bandwidth to load large spatial feature tensors, for example, of 256×64×64 for SAM. We pre-compute the features from off-the-shelf models instead of conducting inference on the fly. This approach requires extra storage space to save all the features extracted from the above foundation models, but significantly saves on training time and avoids loading models with high GPU memory usage during training. All training configurations are listed in Table 11.

Table 11: DreamVLA Training Configuration

| Hyperparameters | Value |
| --- | --- |
| # GPUs | 8 |
| Batch size | 8 / GPU (64 effective) |
| Learning rate (LR) | 1e-3 |
| LR Schedule | Constant |
| Weight decay | 0.01 |
| Optimizer | AdamW |
| Betas | [0.9, 0.999] |
| Epochs | 20 |
| Warm-up epochs | 1 |
| Warm-up LR schedule | Linear (1e-2 * LR) |

## B Experiments

### B.1 Simulation Benchmark and Settings

We evaluate DreamVLA on the CALVIN benchmark [119], a simulated robotic manipulation suite designed for studying long-horizon, language-conditioned tasks. CALVIN aims to facilitate the development of agents that operate solely based on onboard sensor inputs and free-form human instructions, without access to privileged information or external supervision. The tasks in CALVIN require agents to execute long sequences of low-level control commands in response to complex language goals, reflecting realistic robotic interaction scenarios.

The benchmark includes four structurally similar but visually distinct environments, referred to as Env A, B, C, and D. Each environment features a Franka Emika Panda arm with a parallel gripper and a tabletop workspace containing manipulable elements such as a sliding door, a drawer, and a light button. The textures, object placements, and scene layouts vary across environments to encourage generalization and robustness.

Observations consist of RGB images from both fixed and gripper-mounted cameras (resized to 224×224), as well as low-dimensional robot state inputs that include the end-effector's position, orientation, and gripper status. The agent outputs a 7-dimensional continuous action vector: 6 dimensions control the spatial displacement of the gripper, and the final dimension governs the open/close state of the gripper.

The dataset contains approximately 2.4 million interaction steps and 40 million short-horizon action windows. Environments A, B, and C provide language-free demonstrations for large-scale pretraining, while annotated instructions are available in a subset of the data for downstream policy learning. We hold out Env D for evaluation to assess zero-shot generalization to unseen combinations of instructions and environment variations.

Following standard protocol [119, 58], we evaluate performance on a set of 34 diverse tasks that include object pushing, placing, rotating, and other dexterous operations. In contrast to prior work, DreamVLA not only predicts

actions conditioned on visual-language observations but also simultaneously learns to infer comprehensive future world knowledge, including depth maps, dynamic saliency regions, DINOv2 features, and SAM-based segmentation maps. This multi-task supervision enables richer scene understanding and improves policy generalization. We report success rate (**SR**) as our primary evaluation metric, measuring whether the instructed task was completed correctly based on the final state of the environment.

## B.2 Simulation Results

We evaluate our approach on the CALVIN ABC-D benchmark, where training is conducted on environments A, B, and C, and testing is performed exclusively in Environment D. This evaluation setting poses a strong challenge for generalization, as Environment D features novel textures, object arrangements, and visual configurations not seen during training. As reported in Table 1 in the main manuscript, DreamVLA achieves superior performance across all tasks, substantially outperforming previous state-of-the-art methods.

In particular, our model significantly outperforms two-stage inverse dynamics approaches such as Susie [120], demonstrating the effectiveness of our end-to-end architecture that unifies multimodal prediction and action generation. Compared to CLOVER [123], UP-VLA [59], Seer [58], which also incorporates visual foresight, DreamVLA benefits from a more integrated design and joint optimization, resulting in consistently stronger execution accuracy. Furthermore, our method surpasses video generation-based pretraining approaches like GR-1 [16], highlighting the advantage of coupling vision prediction with action planning in a single framework.

Notably, DreamVLA, achieves an average episode length of **4.44** on the ABC-D split, establishing a new state-of-the-art on the CALVIN benchmark and validating the benefits of predicting future knowledge. The qualitative results as shown in Figure 7.

## B.3 Visualization

As shown in Figure 8 and Figure 9, we visualize the model's predictions of dynamic regions and depth maps. Although supervision is applied only to dynamic regions, DreamVLA is able to reconstruct semantically meaningful representations of the entire scene. This surprising generalization ability can be attributed to two factors. First, in long-horizon manipulation sequences, the robot arm is in constant motion and frequently interacts with various objects, causing most task-relevant regions to become dynamic at some point in time. This ensures that a large portion of the scene is eventually observed under dynamic supervision. Second, although static regions are not explicitly supervised, the input frames inherently contain global visual context—including background structures, object appearances, and spatial layout—which the model can leverage to hallucinate and complete missing details. As a result, DreamVLA implicitly learns to integrate temporal dynamics with static priors, leading to coherent and accurate predictions beyond the explicitly labeled regions.

Although the predicted depth maps are relatively coarse due to the patch-level reconstruction inherent in MAE-style decoders [106], they still provide valuable guidance for downstream tasks. In particular, the model benefits from anticipating future depth, which helps refine action decisions and improves spatial awareness.

## B.4 Additional Ablation Study

**Q6: Effect of the query count per modality inside `<dream>` queries.**

Each `<dream>` query contains three groups of elements: dynamic, depth, and semantics, each assigned $K$ queries. We vary $K \in \{4, 9, 16\}$ to examine its influence. When $K = 4$, the limited capacity prevents the model from encoding fine-grained motion, geometry, and semantics, so accuracy drops even though memory usage is lowest. With $K = 9$, each modality has sufficient bandwidth without overload-

Table 12: **Performance comparison** between different numbers of `<dream>` queries.

| Number | Task completed in a row | | | | | |
|---|---|---|---|---|---|---|
| | 1 | 2 | 3 | 4 | 5 | Avg. Len. |
| 4 | 97.2 | 92.6 | 86.4 | 80.7 | 75.1 | 4.32 |
| 9 | **98.2** | **94.6** | **89.5** | **83.4** | **78.1** | **4.44** |
| 16 | 98.1 | 93.0 | 86.9 | 81.0 | 73.9 | 4.33 |

ing the backbone, yielding the best success rate and the longest uninterrupted task execution. Increasing to $K = 16$ introduces redundant tokens that compete for attention and raise GPU memory, bringing no extra gain and slightly lower generalization.

## B.5 Real-world Settings

In our real-world training setup, we use a history length of 7, with the model jointly predicting the next 3 future visual representations and action steps. The visual backbone is initialized with a ViT-B model pre-trained using MAE [106], and inference is accelerated using bfloat16 mixed-precision without any observed degradation in

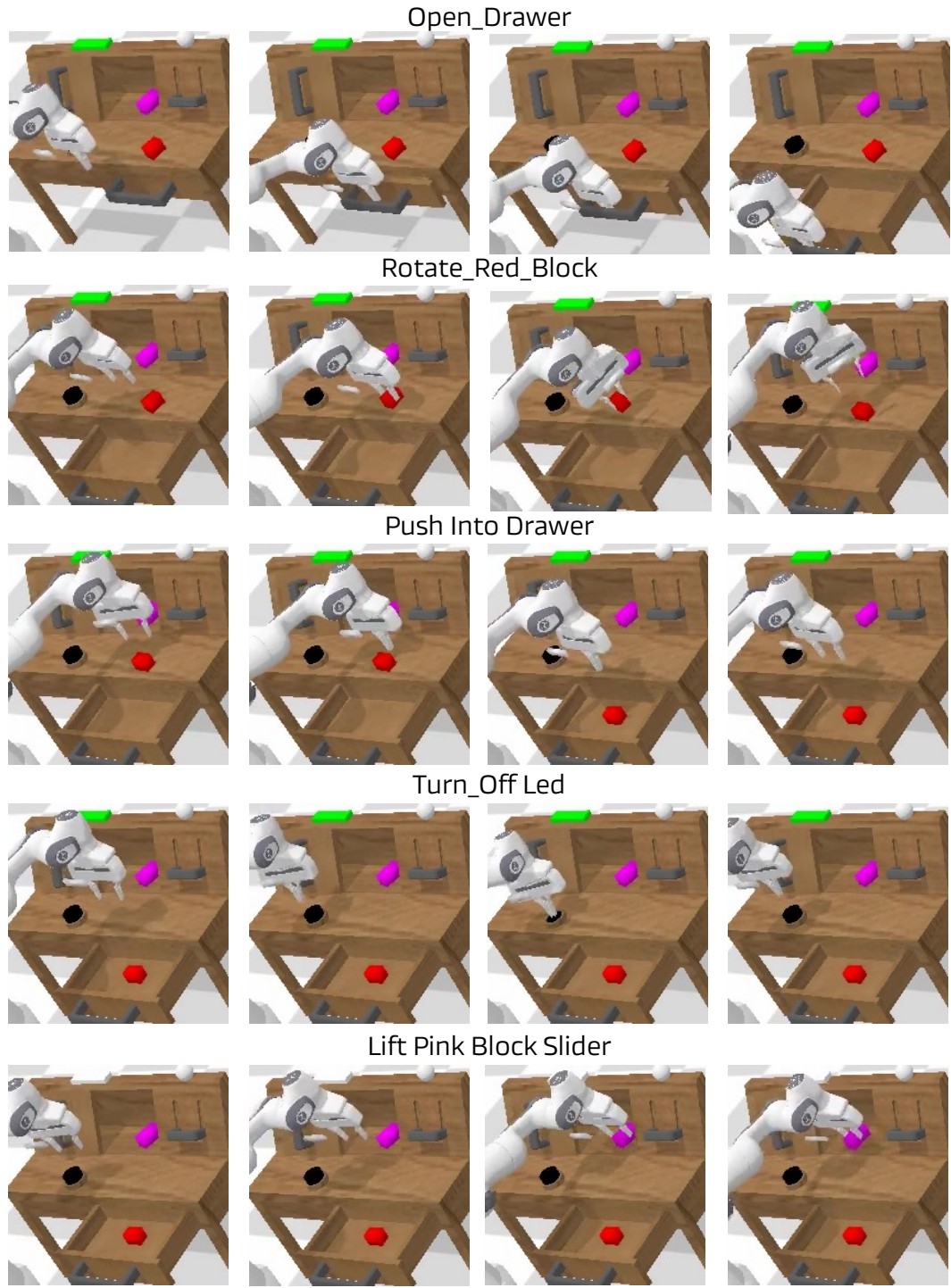

Figure 7: **Qualitative results** of the CALVIN long horizon task.

task performance. This configuration strikes a balance between computational efficiency and policy stability in manipulation tasks.

For pretraining, we leverage a large-scale dataset such as DROID [84], which contains approximately 76,000 successful robot trajectories collected in diverse settings. For downstream adaptation, we fine-tune the model

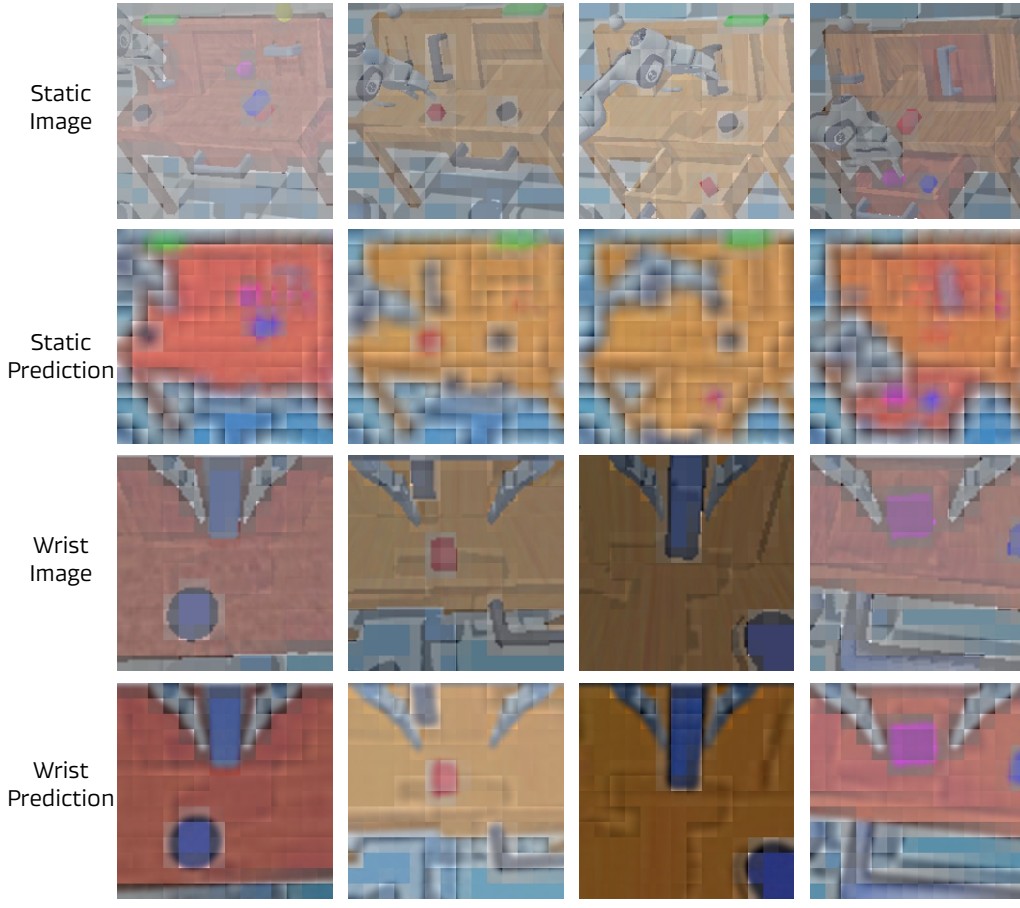

Figure 8: **Visualization results** of the dynamic region predictions.

using 100 task-specific demonstrations for each task collected with SoFar [24]. As shown in Figure 10, we present the qualitative results of real-world experiments.

## B.6 Inference latency

| model part | inference time |
|---|---|
| image, text and state encoders | 12 ms |
| observation forward pass w/dream query | 19 ms |
| w/o dream query | 16 ms |
| action forward pass (10 step) | 60 ms |
| total | 91 ms |
| w/o dream query | 88 ms |

Table 13: Inference time of our model on a NVIDIA GeForce RTX 4090 GPU, we test 5 times and take average time.

Table 13 reports end-to-end latency for processing two camera images on an NVIDIA GeForce RTX 4090. At inference time, no explicit image decoding is required, and the system runs at 11 Hz. The results show: (i) Auxiliary cues incur minimal overhead. Our "dream queries" are token-level predictions (no explicit image decoding and no external models). The incremental cost is 3 ms (3.4%), i.e., 91 ms vs. 88 ms without dream queries. (ii) Latency is dominated by the action head rather than the auxiliary cues. A 10-step action head contributes about 60 ms. This cost scales with the number of sampling steps and model size; it can be reduced by using fewer steps, a smaller DiT variant, faster samplers. (iii) For latency-critical applications, we can prune

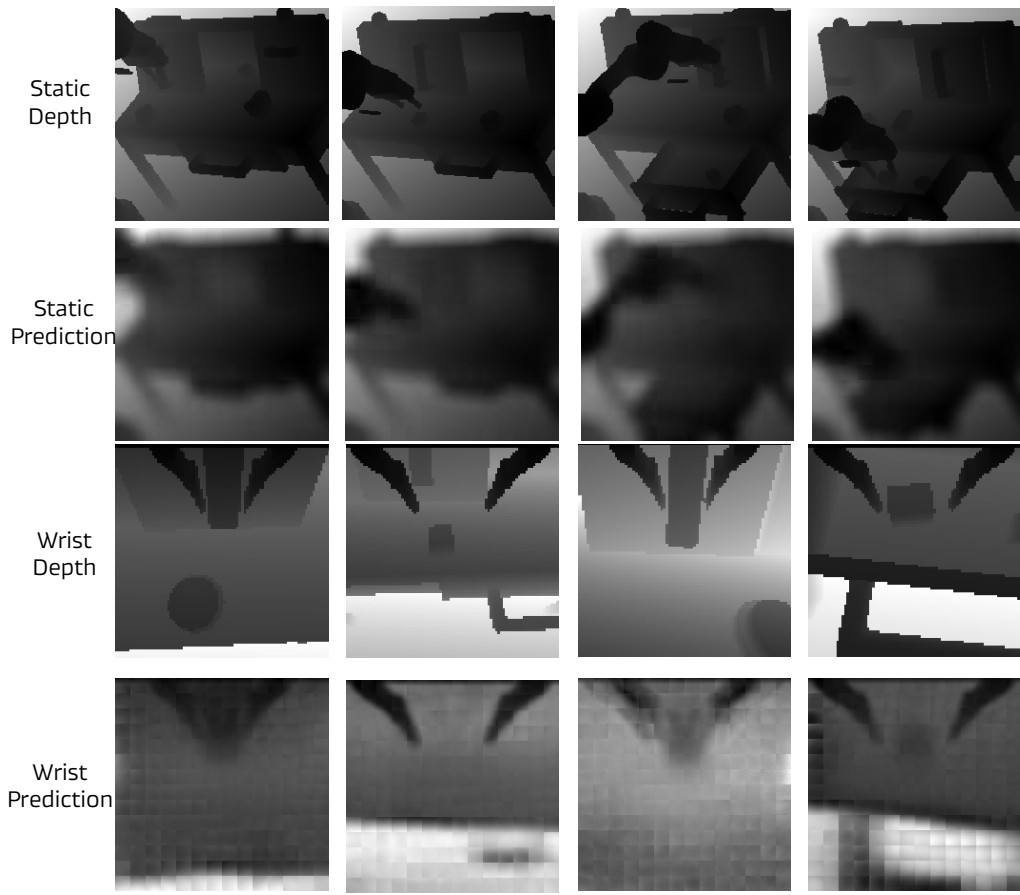

Figure 9: **Visualization results** of the depth maps.

dream queries (e.g., keep only dynamic regions, or dynamic+depth) and/or increase action chunking or run the action head asynchronously to amortize computation, without changing the observation pathway.

## C   Additional Related Works

### C.1   Language-Grounded Robot Manipulation

Language-grounded robot Manipulation adopts the human language as a general instruction interface. Existing works can be categorized into two groups: **i)** *End-to-end* models like RT-series [2, 86, 87] built upon unified cross-modal Transformers with tokenized actions [74, 126–128, 32, 129], large vision-language-action (VLA) models built from VLMs [1], or 3D representations [130, 46, 131]. Training on robot data such as Open X-Embodiment [14] and DROID [84], a remarkable process has been made. However, the data scale is still limited compared to in-the-wild data for training VLMs. **ii)** *Decoupled* high-level reasoning and low-level actions in large vision-language models and small off-the-shelf policy models, primitives [132–138, 24], or articulated priors [139, 140].

## D   Limitation & Future Works

While DreamVLA demonstrates solid vision-language-action and achieves state-of-the-art performance on CALVIN [119], its current scope is still narrow: it practises mainly parallel-gripper manipulation, relies on RGB-centric data, and is trained on scenes with limited geometric and material diversity. We therefore plan to (i) add dexterous-hand demonstrations with rich contact annotations [141, 142], (ii) introduce 3D point clouds [143, 144, 104, 68, 145, 146, 67, 147] and spatial information [24, 148], tactile—and fuse them into volumetric world states, and (iii) extend data collection and on-policy fine-tuning to bolster generalization and long-horizon robustness.

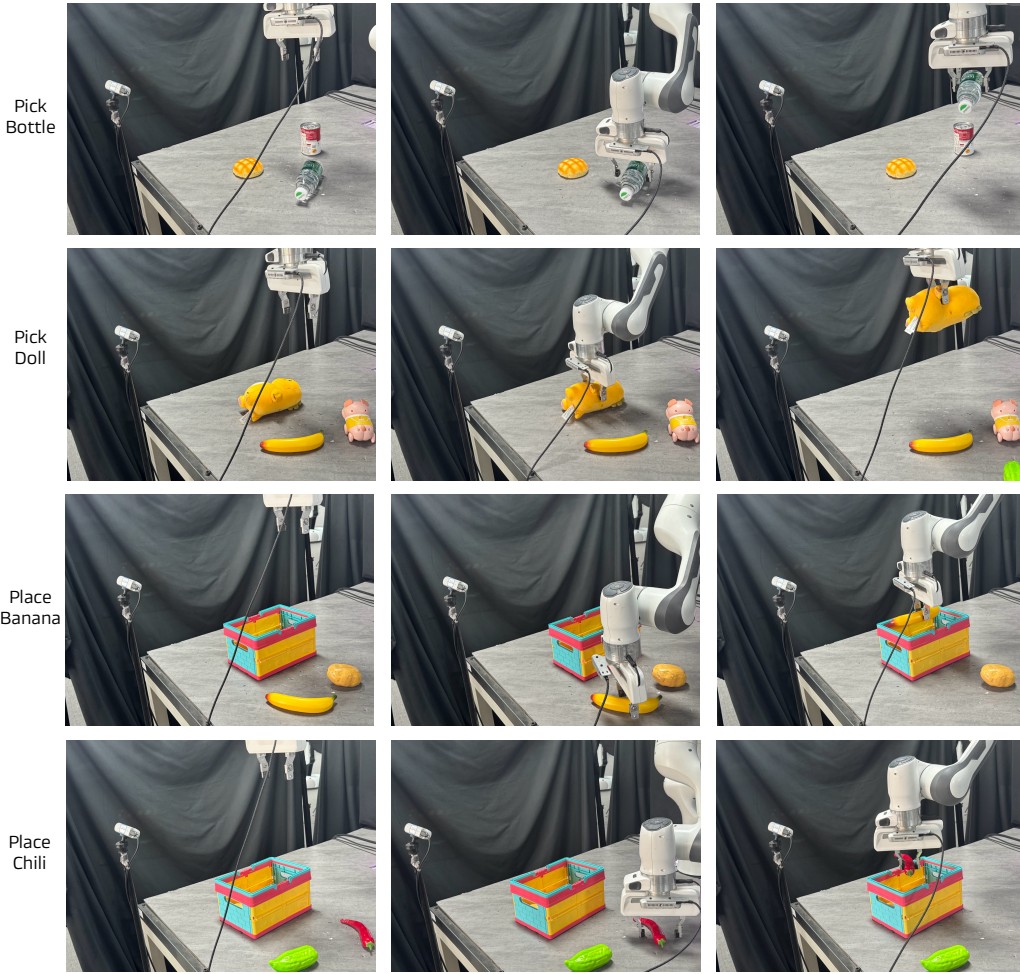

Figure 10: **Qualitative results** of real world language-grounded manipulation.

# E    Additional Discussions and Future Work

**i. Scaling Laws.** A promising direction for future exploration involves investigating scaling behavior in DreamVLA. In particular, we plan to study how increasing the capacity of key components—such as the backbone visual encoder or the size of the language model—affects model performance. This includes replacing the current text encoder with larger-scale language models (e.g., LLaMA-2 or GPT variants) to assess the impact of richer linguistic understanding on multimodal reasoning and action generation.

**ii. Integration with Additional Baselines.** We also aim to evaluate DreamVLA in conjunction with more recent and diverse baselines. For example, RoboVLMs [39] incorporate a wide range of vision-language backbones and offer a unified framework for robotic policy learning. Combining DreamVLA with these baselines can help standardize performance comparisons and reveal architectural synergies between pretrained vision-language models and action-centric transformers.

**iii. Contribution of Multi-View Observations.** Our current framework leverages both fixed and egocentric camera views. In future work, we plan to conduct a detailed ablation study to quantify the contribution of each view modality to task performance. This analysis will provide insights into how multi-view information improves spatial reasoning and robustness, especially in occluded or ambiguous scenarios.

**iv. Extension to More Complex and Long-Horizon Tasks.** While DreamVLA demonstrates strong performance on the CALVIN benchmark, we are interested in extending the framework to more complex, long-horizon tasks that involve extended temporal dependencies, delayed rewards, and multi-stage subgoals. This includes evaluating on benchmarks that require sustained interaction, sequential tool use, or high-level planning. Addressing these challenges will require not only more powerful temporal modeling but also better integration of memory, goal abstraction, and hierarchical reasoning mechanisms.

**v. Application to Robotic Navigation and Humanoid.** Beyond tabletop manipulation, DreamVLA could be adapted to robot navigation tasks in indoor or semi-structured environments. By learning to predict dynamic regions, obstacles, and semantic scene components, the model could support instruction-driven navigation and path planning under multimodal supervision, especially in settings where map-based planning is infeasible.

Furthermore, another compelling extension is applying DreamVLA to humanoid robots, which require reasoning over whole-body motion, balance, and physically grounded interactions. The modularity of our framework allows for integration with additional proprioceptive inputs and more complex action spaces. This line of work would explore how multimodal predictive learning can scale to full-body motor control and human-like task execution.

# F  Broader Impacts

DreamVLA proposes a new training paradigm for vision-language-action (VLA) modeling, going beyond the conventional mapping from visual observations and language to actions. Instead of directly predicting actions from high-dimensional input, our framework first encourages the model to predict comprehensive world knowledge, including depth, dynamic motion, segmentation, and semantic features, before generating actions. This intermediate representation improves action grounding and generalization.

A key strength of DreamVLA lies in its simplicity and efficiency: by adding only a lightweight decoder and a set of learnable queries, we significantly enhance the performance of existing VLA backbones with minimal parameter overhead. This makes the method both scalable and compatible with current VLM-based architectures, paving the way for more robust and transferable policies.

Practically, this design can benefit the development of assistive robots' navigation and humanoid robots, where it is essential for agents to generalize across novel environments and language goals. Furthermore, since our method leverages unlabeled perceptual signals during training, it reduces reliance on curated language-instruction datasets, which are often expensive and domain-specific.

Overall, DreamVLA offers a practical, extensible, and training-efficient framework for improving VLA systems, and we hope it inspires further research into multimodal abstraction and low-cost robot learning.

