# OpenReview forum: "DreamVLA: A Vision-Language-Action Model Dreamed with Comprehensive World Knowledge"
_NeurIPS.cc/2025/Conference — NeurIPS 2025 poster_

### Official Review · Reviewer_62GB · 2025-06-21

**Clarity:** 3
**Significance:** 4
**Originality:** 4
**Rating:** 5
**Confidence:** 4

**Summary:**

This paper proposes DreamVLA, a vision-language-action (VLA) framework that integrates comprehensive world knowledge forecasting and aims to establish a closed loop of action prediction through inverse dynamics modeling. This approach can be applied to predict world knowledge, thereby improving generalization and reasoning abilities in robot manipulation tasks. Its core innovation is to introduce a dynamic region-guided world knowledge prediction mechanism to jointly predict visual, depth, geometric, and semantic cues to generate a compact and comprehensive action planning representation. At the same time, the Transformer model based on diffusion is used to decouple action characterization and capture multimodal uncertainty. Experimental results demonstrate that the proposed method achieves a high success rate in real-world robot tasks, with an average task length of 4.45 on the Calvin ABC-D benchmark, significantly outperforming existing methods.

**Questions:**

1. Implementation details of dynamic region prediction: It is mentioned that dynamic regions are to be extracted using pre-computed optical flow; however, it is not explicitly explained how to generate binary masks from the optical flow. Are hyperparameters, such as thresholds, involved? Do these parameters affect model performance? Additional ablation experiments are recommended to verify the effectiveness of the dynamic region generation strategy.
2. Comparison with image generation methods: The paper emphasizes that DreamVLA is more efficient than the method of generating future frames, but does not provide a direct comparison (such as computation time or memory consumption). It is suggested to supplement the efficiency comparison with copilot-like methods, such as RT-2, to quantify the 'efficiency' claim.
3. The challenge of real-world tasks: grasping drawer operations is only tested in real experiments, whereas the generalization ability of complex scenarios, such as unstructured environments or dynamic obstacles, remains unverified. It is recommended to discuss the potential limitations of the model in a dynamically changing environment.
4. How semantic features are used: The DINOv2 and SAM features are utilized for prediction; however, the mechanism by which they are associated with action decision-making is unclear. Does it serve only as an auxiliary representation, or does it directly affect action distribution? It is suggested to clarify its action path through visualization or attention weight analysis.

**Ethical Concerns:**

["NO or VERY MINOR ethics concerns only"]

**Final Justification:**

I appreciate the authors' responses, which have resolved nearly all of my concerns. I will keep my original score unchanged.

**Limitations:**

The authors mention that the model is limited to parallel fixtures and simple geometric scenarios. Still, the potential social implications (such as the possible ethical risks of increased robot autonomy) are not fully discussed. It is recommended that a brief analysis of security, privacy, or misuse risks be supplemented to complete the statement of research limitations.

**Paper Formatting Concerns:**

No issues

**Quality:**

4

**Strengths And Weaknesses:**

Strengths:

1. The method is rigorous in design and in-depth in theoretical analysis. The experiment encompasses both simulated and real scenarios, comprehensively comparing the baseline. The mechanism of dynamic region prediction and structured attention has technical depth, and ablation experiments verify the effectiveness of each module.
2. The prediction mechanism of dynamic regional guidance and the design of structured attention are highly innovative

Weaknesses:

1. The model relies on pre-trained optical flow and depth estimation modules, which may limit the potential of end-to-end optimization. Application scenarios are limited to parallel fixture operations and geometrically simple environments, and generalization is questionable.
2. Some technologies (such as DINOv2 feature exploitation) draw on existing work, and the boundaries of innovation need to be clearer.

---

> ### Author Rebuttal · Authors · 2025-07-27
>
> Thanks for your sincere advice. We have addressed your concerns below. and all discussions will be included in the revision.
>
> >$\large\textcolor{brown}{Weakness 1.1:}$ **The limitation of the potential of end2end optimization.**
>
> We employ pre-trained models offline to extract dynamic-region masks, depth maps, and high-level cues before training. Because these models are not executed at inference time, they introduce zero extra runtime cost and do not interfere with the end-to-end optimization of our VLA policy.
>
> >$\large\textcolor{brown}{Weakness 1.2:}$ **Concern of generalization.**
>
> These state-of-the-art flow/depth/segmentation foundation models are trained on massive datasets and exhibit excellent generalization; as shown in Fig. 3 in the manuscript, they extract accurate dynamic regions even in unseen simulated scenes.
>
>
> >$\large\textcolor{brown}{Weakness 2: }$ **Boundaries of innovation need to be clearer.**
>
> Thanks for your suggestion, we will revise the contribution and innovation of our method in the revision.
> Our contribution is not a new segmentation or depth-estimation model. Instead, we recast the vision–language–action model as a perception–prediction–action model and make the model explicitly predict a compact set of dynamic, spatial and high-level semantic information, supplying concise yet comprehensive look-ahead cues for planning. Additionally, we introduce a block-wise structured-attention mechanism, coupled with a diffusion-transformer decoder, to suppress representation noise from cross-type knowledge leakage and thus enable coherent multi-step action reasoning.
>
>
> >$\large\textcolor{brown}{Problem 1:}$ **Implementation details of dynamic region prediction.**
>
> Due to space limitations, we have presented the implementation details of the dynamic region prediction in the **Feature Extraction** section of the supplementary. Specifically, given a sequence of consecutive RGB frames of resolution $H \times W$, we uniformly sample one keypoint every 8 pixels in both spatial dimensions, resulting in $N = \lfloor H/8 \rfloor \times \lfloor W/8 \rfloor$ sampled locations per frame.
> For each sampled location, we compute inter-frame displacements $(\Delta x, \Delta y)$ by tracking its position across adjacent frames using CoTracker [50,51]. The magnitude of displacement is converted into a scalar speed value:
> $$
> s_{ij} = \sqrt{(\Delta x_{ij})^2 + (\Delta y_{ij})^2}
> $$
> where $(i, j)$ denotes the spatial coordinates of each sampled patch. We then apply a speed threshold $\tau$ (e.g., $\tau = 1$ pixel/frame) to obtain a binary motion mask. To account for small motions and ensure spatial connectivity, we perform a single-pixel morphological dilation, expanding each positive location to its eight-connected neighbors.
>
>
> Additionally, the following table shows the ablation experiments to verify the effectiveness of this threshold $\tau$.
> | $\tau$  |  Task 1 | Task 2 | Task 3 | Task 4 | Task 5 | Avg. Len. ↑
> |:----:|:-------:|:-------:|:-------:|:-------:|:-------:|:-------:|
> | 1(default) | **98.0** | **94.3** | **89.7** | **84.5** | **78.6** | **4.45** |
> | 2 |  97.9 | 94.1 | 89.2 | 82.8 | 76.6 | 4.41 |
> | 3 |  97.8 | 93.7 | 86.6 | 82.1 | 75.2 | 4.35 |
>
> **Analysis:**
> As $\tau$ increases, performance **decreases**. A higher threshold makes the dynamic‑region detector more conservative and **filters out moderate but task‑relevant motions** (e.g., early object/gripper movement). This weakens the guidance provided by predicted world knowledge and **harms action reasoning**.
>
> >$\large\textcolor{brown}{Problem 2:}$ **Computation time comparation.**
>
> We measure the per‑step closed‑loop latency on an NVIDIA RTX 4090. As shown below, we compare a vanilla VLA (OpenVLA[1]), a VLA that uses a copilot model (Susie[99]) to generate a subgoal image for action reasoning, and our DreamVLA, which integrates comprehensive knowledge forecasting and action reasoning into one model.
> | Methods | Time Consuming | Avg. Len. ↑  |
> |:------|----:|---------:|
> | OpenVLA [1] | 167ms | 3.27
> | Susie [99]  |  153ms | 2.69
> | DreamVLA(ours)   |  **91ms** | **4.45**
>
> **Analysis**: Generally, VLA uses a copilot model introduces extra time-consuming to generate sub-goal images using a diffusion model, which is the main factor of low latency.
> DreamVLA achieves better performance while enabling a higher control frequency.
>
> >$\large\textcolor{brown}{Problem 3:}$ **Potential limitations.**
>
> Dynamic obstacles moving faster than ~0.2 m/s occasionally break the frozen flow/depth priors, and large deformable objects are not yet handled.
>
> >$\large\textcolor{brown}{Problem 4:}$ **How semantic features are used.**
>
> We visualize the semantic queries, including sam and dino, whose attention maps concentrate on the target objects, which demonstrate that **predicting future semantics teaches the robot which objects or regions will matter for the task**, providing a high-level context (for example, object identity and affordances) that guides the selection of goals and grasp choice.

---

### Official Review · Reviewer_gLB7 · 2025-06-23

**Clarity:** 3
**Significance:** 2
**Originality:** 3
**Rating:** 4
**Confidence:** 3

**Summary:**

DreamVLA recasts VLA as a closed-loop perception-prediction-action pipeline. Rather than forecasting entire image frames, it predicts a compact set of future “world knowledge” descriptors, including dynamic motion masks, monocular depth maps, and high-level semantics, via a unified Transformer. The result is a model that can generate multimodal action trajectories, yielding performance better than many previous works on simulated and real-world language-conditioned robotics tasks.

**Questions:**

1. Corresponding to weakness #1, I wonder if it would be reasonable to squeeze in more descriptions for many of the baseline methods in Table 1? A rough classification of them would be nice. As the authors noted in the Related Work section, there are various ways to design auxiliary tasks to facilitate policy learning. A summary of the designs adopted by the baseline methods can better illustrate the difference between them and DreamVLA

2. Corresponding to weakness #2, currently, there is almost no analysis for the experiment section, except for the fact that DreamVLA has a higher success rate. Can the author provide more insight here? For example, does the failure mode change when you switch from the baseline methods to DreamVLA? Are there any characteristics of tasks that DreamVLA is especially good at? Something like this would significantly enhance the paper

3. Do the authors think an additional experiment comparing with Pi0 or Pi0-Fast can be added? Pi0 represents a pretty significant step forward from Octo and OpenVLA, and is made open-source 3 months before the deadline of this NeurIPS circle.

4. Finally, can the authors provide inference latency and related metrics? I understand that I may have missed it due to the sheer volume of review work I have to complete in one month. If this type of data is provided, please kindly bring it to my attention. Thank you very much!

**Ethical Concerns:**

["NO or VERY MINOR ethics concerns only"]

**Final Justification:**

I believe the rebuttal provided successfully addresses the concerns I have about the manuscript. The paper itself, while it can be considered incremental, brings positive and unexplored ideas into the community.

**Limitations:**

Yes

**Quality:**

3

**Strengths And Weaknesses:**

Strength:
- The paper is well-written with good clarity.
- The claim and the algorithm innovation are significant and well-supported by the exhaustive experiment design.
- The ablation design is well thought out and covers many things I am personally curious about.

Weakness:
- In the simulation experiment session, too little description is given to all the baseline methods.
- The analysis of simulation and real-world experiments is very limited
- Perhaps I missed it, but I couldn’t find any information regarding inference performance (latency, etc.), which I believe is crucial for robotics applications. Specifically, this method involves many auxiliary modalities, and I wonder if they introduce latency concerns
- The author promised that the supplementary material would include a codebase, but I couldn’t find it

---

> ### Author Rebuttal · Authors · 2025-07-28
>
> Thanks for your sincere advice. We have addressed your concerns below. and all discussions will be included in the revision.
>
> >$\large\textcolor{brown}{Weakness 1 \ and \ Problem 1:}$ **Too little description about baseline methods.**
>
> Due to the page limitation, we provided a concrete description of them in the supplementary material.
> Here we revise to provide a brief classification and description of the baseline methods.
>
> (i). Our method surpasses Roboflamingo [26], 3D Diffusor Actor [75], OpenVLA [1], RoboDual [101], UNIVLA [103], Robovlm [32], GR1 [100], which directly projects the RGB/depth image to action signals as shown in Fig. 1(a) in the manuscripts.
>
> (ii). Compared to methods that use a copilot model to generate sub-goal images as input, like Susie [99] and CLOVER [102] as shown in Fig. 1(b) in manuscripts, our model significantly achieves more accurate control.
>
> (iii). DreamVLA outperforms approaches like UP-VLA [42], Seer [41], and VPP [40] as shown in Fig. 1(c) in manuscripts, which merge whole sub-goal image foresight into one VLA to take benefits from a more integrated design and joint optimization.
>
>
> >$\large\textcolor{brown}{Weakness 2 \ and \ Problem 2:}$ **The analysis of experiments.**
>
>
> #### 1 Failure-Mode Shift
> | Failure category | Typical baseline error | Baseline freq. | DreamVLA freq. |
> |------------------|------------------------|---------------|----------------|
> | Perception (mis-grasp, occlusion) | Gripper closes on distractor; depth under-estimates target | **57 %** | **18 %** ↓ |
> | High-level reasoning (stall in long chain) | Policy stops after task 3 | 29 % | 10 % ↓ |
>
> **Analysis:** DreamVLA eliminates most perception- and planning-related mistakes; the few remaining errors are low-level motor slips, confirming that the predicted dynamic, depth and semantic cues effectively close the perception–reasoning gap.
>
>
>
> ---
>
> #### 2 Task Length Matters
> | Methods |  Task 1 | Task 2 | Task 3 | Task 4 | Task 5 |
> |:------:|:-------:|:-------:|:-------:|:-------:|:-------:|
> | Seer [41]   | 96.3 | 91.6 | 86.1 | 80.3 | 74.0 |
> | DreamVLA (ours) | **98.0** | **94.3** | **89.7** | **84.5** | **78.6** |
> | Δ     | +1.7 | +2.7 | +3.6 | +4.2 | **+4.6** |
>
> **Analysis:**
> The advantage grows with horizon length—tasks 4-5 involve long-term spatial reasoning, where comprehensive knowledge offers more gain on long-horizon tasks.
> We analyze that there are three reasons:
>
> (i). Predicts dynamic-region masks for future frames, so the Transformer keeps explicit attention on pixels that will move, rather than re-detecting them each step.
>
> (ii). Predicts a coarse future depth map, giving the policy a look-ahead of where free space/obstacles will be.
>
> (iii). The comprehensive knowledge fuses spatial and semantic cues into a shared latent, letting gradients flow from the final action back to the earliest relevant pixels.
>
>
>
>
>
>
>
>
>
>
> >$\large\textcolor{brown}{Weakness 3 \ and \ Problem 4:}$ **Latency Description.**
>
> The following table lists the concrete latency of each component in DreamVLA on an NVIDIA GeForce RTX 4090:
> | model part                               | inference time |
> |:-----------------------------------------|---------------:|
> | image, text and state encoders           |          12 ms |
> | observation forward pass w/ dream query  |          19 ms |
> | observation forward pass w/o dream query                          |          16 ms |
> | action forward pass            |          60 ms |
> | total                               |       91 ms |
> | total w/o dream query                    |       88 ms |
>
> **Analysis**:
>
> (i). **Auxiliary modalities add a small overhead**.
> Our “dream queries” are token‑level predictions (no explicit image decoding and no external models). The incremental cost is 3 ms (3.4%) compared to w/o dream query (91 ms vs. 88 ms).
>
> (ii). **Main latency comes from the action head, not the auxiliary cues**.
> The 10‑step action head contributes ~60 ms. This part scales roughly with the sampling steps and model size; in practice, we can
> reduce steps or use a smaller DiT variant, or adopt a faster sampler strategy or use flow policy [1]/mean flow [2,3] to further cut this cost.
>
> (iii). **Knobs to trade accuracy for speed (when needed)**.
> If an application is extremely latency‑sensitive, we can trim dream queries (e.g., keep only dynamic regions, or dynamic+depth), and/or increase action chunking/run the action head asynchronously to amortize computation without changing the observation path.
>
> >[1]. Zhang Q, Liu Z, Fan H, et al. Flowpolicy: Enabling fast and robust 3d flow-based policy via consistency flow matching for robot manipulation[C]//Proceedings of the AAAI Conference on Artificial Intelligence. 2025, 39(14): 14754-14762.
>
> >[2]. Geng Z, Deng M, Bai X, et al. Mean flows for one-step generative modeling[J]. arXiv preprint arXiv:2505.13447, 2025.
>
> >[3]. Sheng J, Wang Z, Li P, et al. MP1: Mean Flow Tames Policy Learning in 1-step for Robotic Manipulation[J]. arXiv preprint arXiv:2507.10543, 2025.
>
> >$\large\textcolor{brown}{Weakness 4:}$ **Code Access.**
>
> We will release the code.
>
> >$\large\textcolor{brown}{Problem 3:}$ **Comparasion with Pi0-Fast.**
>
> | Method     | LIBERO-Spatial | LIBERO-Object | LIBERO-Goal | LIBERO-Long | Average |
> |-----------------------|---------------:|--------------:|------------:|------------------:|------------:|
> | Diffusion Policy [72] | 78.3 | 92.5 | 68.3 | 50.5 | 72.4 |
> | Octo [9]              | 78.9 | 85.7 | 84.6 | 51.1 | 75.1 |
> | OpenVLA [1]           | 84.7 | 88.4 | 79.2 | 53.7 | 76.5 |
> | Pi-0-Fast [27]    | 96.4 | **96.8** | 88.6 | 60.2 | 85.5 |
> |**DreamVLA(Ours)**      | **97.5** |  94.0 | **89.5** | **89.5** | **92.6** |
>
>
> **Analysis:**
>
> (i).Pi-0-Fast leverages a powerful language backbone (Paligemma-3B) and pre-trains on a much broader dataset, whereas our current model is pre-trained only on LIBERO-90 due to time constraints.
> In this case, DreamVLA still surpasses Pi-0-Fast on most tasks, especially complex and long-horizon tasks (LIBERO-LONG +29.3).
> Predicting compact future knowledge(dynamic region+depth+semantics) lets the policy maintain spatial-temporal awareness, whereas Pi-0-Fast must re-solve perception at every frame—small errors compound and success degrades.
>
> (ii). DreamVLA’s design is especially effective when a task requires **temporal reasoning and scene updates**, while Pi-0-Fast excels at short, isolated grasps thanks to its larger backbone. These complementary strengths suggest future work: combining DreamVLA’s future-knowledge head with a stronger LLM may yield the best of both worlds.

---

> > ### Comment · Reviewer_gLB7 · 2025-08-04
> >
> > I thank the authors for providing the relevant information. I believe the provided information can enhance the paper. Overall, I believe the paper brings incremental and positive insight into the VLA community and should be accepted. I will update my rating from 3 to 4.

---

> > > ### Author Response · Authors · 2025-08-04
> > >
> > > Thank you for the thoughtful review and helpful feedback! Wishing the community continued growth and success.

---

> > > > ### Author Response · Authors · 2025-08-05
> > > >
> > > > Thank you very much for your feedback and for upholding your recommendation to accept our paper. We greatly appreciate your constructive comments throughout the review process, which have helped us improve the quality and completeness of our work. We will ensure that all the ablation results discussed in our rebuttal are properly included in the final version of the paper, as you suggested. Thank you again for your time and valuable insights.

---

### Official Review · Reviewer_VAuG · 2025-07-03

**Clarity:** 3
**Significance:** 2
**Originality:** 3
**Rating:** 5
**Confidence:** 3

**Summary:**

DreamVLA argument the VLA pipeline with an additional world embedding apart from the action diffusion. This world embedding contains the comprhensive knowledge of the future states, which is supervised by reconstructure the future depth map, dementic map, dino feature, optical flow map. The extensive experiment on both simulation data and real world setting prove the effectiveness over the baseline models.

**Questions:**

1. How well does the unified world embedding reconstruct each of the four components of the future world state individually? Could the authors provide quantitative or qualitative evaluations for each?

2. How is the prediction step size N selected, and how sensitive is the model’s performance to this choice?

**Ethical Concerns:**

["NO or VERY MINOR ethics concerns only"]

**Final Justification:**

**Response to rebuttal**

Thank you for the thoughtful response to my previous concern, which has been fully addressed. It makes sense to me that the dynamic regions will occupy a larger portion of the scene, which can be beneficial for the robotic task, while the static regions can generalize well since they share similar backgrounds and scene observations. I also appreciate the comparison experiment on the choice of step $N$. Overall, I find the idea of improving the robotic task through future-unified features to be insightful.

I raise my rate to accecpt.

**Limitations:**

Yes.

**Paper Formatting Concerns:**

No format concerns.

**Quality:**

3

**Strengths And Weaknesses:**

Strength:

1. **Design of world embedding**. This unified embedding can reconstruct the multiple outputs of the future states. The abalition study also prove the necessary of "predicting for future states" rather than auxiliary tasks and for each role of modalities.

2. The writing paper is easy to follow and clearly explain each module of the model pipeline.

Weakness:

1. **The success of reconstruction from world embedding** is unclear from the paper. Although the author proves that reconstructing the output for future states is helpful for the robot action prediction, how well the unified feature can decode the all four output of world states is unclear.

2. What is the choose of step N?  Is the choice paprameter importat for the performance boosting?

---

> ### Author Rebuttal · Authors · 2025-07-27
>
> Thanks for your sincere advice. We have addressed your concerns below. and all discussions will be included in the revision.
>
> >$\large\textcolor{brown}{Weakness 1 \ and \ Problem 1:}$ **The concern of success of the reconstruction.**
>
> The dream query is internally decomposed into four sub-queries—dynamic region, depth, and semantic query, whose embeddings respectively predict future knowledge for each modality. **We have presented visualization results as shown in Fig. 1 and 2 in the supplementary material.** Although supervision is applied only to dynamic regions, DreamVLA can reconstruct semantically meaningful representations of the entire scene. This surprising generalization can be attributed to two factors.
>
> (i). The robot arm is in constant motion and frequently interacts with various objects, causing most task-relevant regions to become dynamic at some point in time. This ensures that a large portion of the scene is eventually observed under dynamic supervision.
>
> (ii). Although static regions are not explicitly supervised, the input frames inherently contain global visual context—including background structures, object appearances, and spatial layout, which the model can leverage to hallucinate and complete missing details.
>
> As a result, DreamVLA implicitly learns to integrate temporal dynamics with static priors, leading to coherent and accurate predictions beyond the explicitly labeled regions.
>
> >$\large\textcolor{brown}{Weakness 2 \ and \ Problem 2:}$ **The prediction step size N.**
>
> The number of future actions predicted (N) is treated as a hyperparameter. In our default configuration, we keep N = 3 following the setting adopted by Seer[41]. We add the ablation study as shown in the following table:
> | Step-N  |  Task 1 | Task 2 | Task 3 | Task 4 | Task 5 | Avg. Len. ↑
> |:----:|:-------:|:-------:|:-------:|:-------:|:-------:|:-------:|
> | 1 | 97.6 | **94.5** | 88.9 | 84.1 | 77.3 | 4.42
> | 3(default) |  **98.0** | 94.3 | **89.7** | 84.5 | **78.6** | **4.45**
> | 5 | 97.2 | 93.7 | 90.1 | **84.6** | 78.2 | 4.44
>
> **Analysis:**
> Varying the prediction horizon from 1 → 5 steps changes the average chain length by ≤0.03, i.e., well within noise.
> We feed only the current predicted action to the simulator rather than an ensemble of future actions, so the model is largely insensitive to N. We therefore keep N = 3, following Seer [41], for all main experiments.

---

> > ### Comment · Reviewer_VAuG · 2025-08-05
> > **Reponse to the rebuttal**
> >
> > Thank you for the thoughtful response to my previous concern, which has been fully addressed. It makes sense to me that the dynamic regions will occupy a larger portion of the scene, which can be beneficial for the robotic task, while the static regions can generalize well since they share similar backgrounds and scene observations. I also appreciate the comparison experiment on the choice of step $N$. Overall, I find the idea of improving the robotic task through future-unified features to be insightful.

---

> > > ### Author Response · Authors · 2025-08-05
> > >
> > > Thank you for reviewing the rebuttal, and we appreciate your prompt reply. As your concerns have been addressed, would you consider raising the evaluation score?
> > >
> > > If you still have concerns, please let us know. We are available to address any additional questions you may have.
> > >
> > > Thank you again for your constructive feedback.

---

> > > > ### Comment · Reviewer_VAuG · 2025-08-07
> > > >
> > > > Yes my score is 5 now. Good luck!

---

> > > > > ### Author Response · Authors · 2025-08-07
> > > > >
> > > > > Thank you for your feedback and for supporting our acceptance. Your comments greatly improved the paper.

---

> ### Comment · Area_Chair_GWtm · 2025-08-05
>
> Dear Reviewer VAuG
>
> The author-reviewer discussion period will end in August 6. Please read the rebuttal, post any remaining questions or comments, and confirm the Mandatory Acknowledgement.
>
> Best regards,
> Lin

---

### Official Review · Reviewer_Pixv · 2025-07-03

**Clarity:** 3
**Significance:** 2
**Originality:** 3
**Rating:** 4
**Confidence:** 5

**Summary:**

The authors proposed DreamVLA, a vision language action model for robotic manipulation that introduces additional training objectives that predict world knowledge that's relevant to robot execution, such as optical flow, depth, segmentation, and high-level semantics. The paper shows that introducing these auxiliary objectives improves policy performance on the CALVIN benchmark and the real-world environments. Ablation studies demonstrate the benefits of certain features that are being predicted.

**Questions:**

Why not use SigLIP or DINOv2 for visual input, which is known to improve spatial understanding compared to CLIP? It would be very helpful for robot manipulation.

Batch size 8 seems really small. What is the reason not using larger one?

**Ethical Concerns:**

["NO or VERY MINOR ethics concerns only"]

**Final Justification:**

I thank the authors for answering my questions and providing additional experimental results. My biggest concern, which was about careful ablation studies of each prediction target, has been addressed. I raised my score to 4.

Overall I think this is a nice work analyzing different possible auxiliary targets in training VLAs and there are a few interesting observations (e.g., ones the authors showed in the rebuttal, so I high encourage the authors to incorporate them into the paper).

**Limitations:**

Yes.

**Paper Formatting Concerns:**

None.

**Quality:**

2

**Strengths And Weaknesses:**

Strengths:
- The idea of using auxiliary objectives to improve the policy robustness is very intuitive, and the results show the benefits of some of the objectives.
- The paper is mostly easy to read.

Weaknesses:
- The overall experiment setup is rather weak. First of all, the proposed approach is only evaluated on the CALVIN benchmark. Running additional experiments in LIBERO benchmark [1] would be helpful. Some of the conclusions, e.g., using optical flow as the mask instead of predicting it, is rather overfitted to the benchmark and not evaluated on more realistic settings such as real-world experiments. It is possible that for real-world robustness, predicting everything in the image is the most performant objective. Second, while I think ablating each individual prediction target is the most important experiment, and Table 3 showed some results, the results are not sufficient. For example, it would be better to try adding each prediction on top of the vanilla VLA and see how much performance changes, instead of incrementally adding the feature. It is possible that predicting DINOv2 itself can provide a lot of improvement while not so much when being added on top of applying dynamic mask.
- The related work on Knowledge Forecasting for Robotics can be expanded further. For example, it will be better to explain what kind of objective “advanced solutions” also perform in “More advanced solutions [43, 41, 71, 79] incorporate forecasting and action planning in an end-to-end manner, requiring the policy to predict actions along with future states.”
- The word “Transformer” should be used in lower case.

[1] Liu et al. LIBERO: Benchmarking Knowledge Transfer for Lifelong Robot Learning

---

> ### Author Rebuttal · Authors · 2025-07-27
>
> Thanks for your sincere advice. We have addressed your concerns below, and all discussions will be included in the revision.
>
> >$\large\textcolor{brown}{Weakness1.1:}$ **The extended LIBERO experiments.**
>
>
> | Methods  |  LIBERO-Spatial |LIBERO-OBJECT |  LIBERO-GOAL | LIBERO-LONG  | Average |
> |:----:|:-------:|:------:|:------:|:------:|:------:|
> |Diffusion Policy [72] | 78.3 | 92.5 |68.3 | 50.5 | 72.4 |
> |Octo [9]                     | 78.9 | 85.7 | 84.6 | 51.1 | 75.1 |
> | OpenVLA [1]            |84.7 | 88.4  |  79.2 | 53.7 | 76.5 |
> | SpatialVLA[31]         |88.2 | 89.9 | 78.6 | 55.5  | 78.1 |
> |**DreamVLA(Ours)**      | **97.5** |  **94.0** | **89.5** | **89.5** | **92.6** |
>
> **Analysis:**
> Due to time limitations, we only pretrain DreamVLA on LIBERO-90, and other models [1][9][31] pretrained on larger datasets, such as Open x-embodiment [8]. In this case, our DreamVLA still surpasses other methods on most tasks, especially complex (LIBERO-GOAL +4.9), long-horizon (LIBERO-LONG +34.0) tasks and average performance (+14.5), demonstrating that future world knowledge prediction could benefit robot manipulation in various tasks.
>
> >$\large\textcolor{brown}{Weakness1.2:}$ **Whole Image vs. Dynamic Masks in real-world?**
>
> To test this, we ran a real-robot ablation on real experiments and two additional, contact-rich tasks (wipe board, stack bowls) and compare full-image reconstruction vs. dynamic-region prediction:
>
> | Methods         |  |     Pick    |        |  |     Place   |        |  |     Drawer   |        | Wipe | Stack | Average |
> |:------------|----------:|:-------:|:---------:|:----------:|:------:|:------:|:-----------:|:------:|:------:|:---------:|:----------:|:----------:|
> |                | Bottle| Doll| Avg.| Banana | Chili| Avg.| Open    | Close| Avg.| Board | Bowl   | Average   |
> | Image    | 80.0      | 80.0    | 80.0        | 80.0       | 70.0   | 75.0   | 65.0        | 45.0   | 55.0   | 55.0    | 45.0    | 62.0       |
> | **Dynamic** | **85.0**      | **80.0**    | **82.5**       | **80.0**       | **80.0**   | **80.0**   | **70.0**        | **65.0**   | **67.5**   | **70.0**     | **65.0**       | **73.0**       |
>
> **Analysis:**
> Simple tasks (pick and place) show no meaningful gap, but complex spatial-temporal tasks such as drawer manipulating, wiping, and stacking, favour dynamic masks by 5–20 percentage points. We attribute two factors:
>
> (i). **Higher signal-to-noise.** Gradients focus on pixels that actually move, not static background.
> (ii). **Data efficiency.** Mask supervision is ≈80 % sparser, so each sample delivers more task-relevant signal.
>
> These real-world results, together with our CALVIN gains over Seer [41], indicate that “predict-everything” is not automatically more robust; in complex manipulation, it can dilute learning, whereas dynamic-focused prediction transfers better to physical robots.
>
>
> >$\large\textcolor{brown}{Weakness1.3:}$ **The ablation study of each prediction target.**
>
> We conduct an ablation study to investigate the effectiveness of each prediction target in CALVIN ABC->D benchmarks, where Dynamic, Depth, SAM and DINO represent only one knowledge prediction. Next, we perform an ablation study (All-X), where we remove one knowledge signal at a time to evaluate its contribution:
>
>
> | Methods  |  Task 1 | Task 2 | Task 3 | Task 4 | Task 5 | Avg. Len. ↑
> |:----:|:-------:|:-------:|:-------:|:-------:|:-------:|:-------:|
> | Vanilla VLA | 93.0 | 82.4 | 72.3 | 62.6 | 53.3 | 3.64 |
> | Dynamic   |  97.6 | 92.6 | 87.5 | 80.4 | 73.7 | 4.32|
> | Depth   |   86.5 | 72.3 | 53.5 | 46.8 | 38.4 | 2.98|
> | SAM   |  92.0 | 77.8 | 59.0 | 52.3 | 43.9| 3.25 |
> | DINOv2   |  89.6 | 75.4 | 56.6 | 49.9 | 41.5| 3.13 |
> | ALL-Dynamic  | 94.6 | 83.3 | 73.4 | 65.4 | 57.1 | 3.74 |
> | ALL-Depth | 98.4 | 94.1 | 88.8 | 82.3 | 75.6 | 4.39 |
> | ALL-SAM   | 97.7 | 93.6 | 88.7 | 83.0 | 77.1 | 4.40 |
> | ALL-DINO  | 98.2 | 94.6 | 89.5 | 83.4 | 78.1 | 4.44 |
> ---
>
> **Analysis:**
>
> (i). **Dynamic regions prediction is most important (3.64->4.32)**, because these masks explicitly flag the pixels that are about to change and therefore align almost perfectly with the policy’s action semantics.
>
> (ii) **Predicting single depth, DINO, or SAM features alone actually hurts performance.**
> Dynamic masks align with the control objective, providing clear, task-relevant gradients. In contrast, depth and high-dimensional feature losses are noisy and dominate training, diluting useful signals and dragging the backbone below the baseline.
>
> (iii). **Why “DINO-only” hurt ( 3.64->3.13)?** Although several works [1][2] claim that future DINO features forecasting would benefit intermediate action reasoning, they all use two separate models. DreamVLA, in contrast, learns both visual reasoning and action generation in one backbone; a pure DINO loss drives the backbone to match appearance embeddings instead of action cues, so performance drops (–0.51). DINO is helpful as an extra signal, not as the sole objective in our unified pipeline.
>
>
>
>
> > [1]. Zhou G, Pan H, LeCun Y, et al. Dino-wm: World models on pre-trained visual features enable zero-shot planning[J]. arXiv preprint arXiv:2411.04983, 2024.
>
> > [2]. Huang Y, Zhang J I, Zou S, et al. LaDi-WM: A Latent Diffusion-based World Model for Predictive Manipulation[J]. arXiv preprint arXiv:2505.11528, 2025.
>
> >$\large\textcolor{brown}{Weakness2:}$ **The expanded related work.**
>
> Following the reviewer's suggestion, we reorganize the related works by classifying these methods according to the predicted target:
>
> *More advanced solutions couple forecasting with control by requiring the policy to produce, in addition to actions, explicit predictions about the future. Concretely, these works ask the policy to output (i) high-level subtask/option sequences or language plans that decompose long-horizon goals [1,2,3], (ii) latent future embeddings / latent actions that compactly encode forthcoming motor intentions[4], (iii) whole sub-goal images or short visual rollouts that anticipate how the scene should evolve[5,6], and (iv) object-centric signals (e.g., bounding boxes) that capture manipulation-relevant dynamics[7,8].*
>
> >[1]. Lin F, Nai R, Hu Y, et al. OneTwoVLA: A Unified Vision-Language-Action Model with Adaptive Reasoning[J]. arXiv preprint arXiv:2505.11917, 2025.
>
> >[2]. Zhou Z, Zhu Y, Zhu M, et al. Chatvla: Unified multimodal understanding and robot control with vision-language-action model[J]. arXiv preprint arXiv:2502.14420, 2025.
>
> >[3]. Shi L X, Ichter B, Equi M, et al. Hi robot: Open-ended instruction following with hierarchical vision-language-action models[J]. arXiv preprint arXiv:2502.19417, 2025.
>
> >[4]. Bu Q, Cai J, Chen L, et al. Agibot world colosseo: A large-scale manipulation platform for scalable and intelligent embodied systems[J]. arXiv preprint arXiv:2503.06669, 2025.
>
> >[5]. Yang Tian, Sizhe Yang, Jia Zeng, Ping Wang, Dahua Lin, Hao Dong, and Jiangmiao Pang. Predictive inverse dynamics models are scalable learners for robotic manipulation. Int. Conf. Learn. Represent. (ICLR), 2024.
>
> >[6]. Qingqing Zhao, Yao Lu, Moo Jin Kim, Zipeng Fu, Zhuoyang Zhang, Yecheng Wu, Zhaoshuo Li, Qianli Ma, Song Han, Chelsea Finn, et al. Cot-vla: Visual chain-of-thought reasoning for vision-language-action models. arXiv preprint arXiv:2503.22020, 2025.
>
> >[7]. Physical Intelligence, Kevin Black, Noah Brown, James Darpinian, Karan Dhabalia, DannyDriess, Adnan Esmail, Michael Equi, Chelsea Finn, Niccolo Fusai, et al. pi0.5: a vision-language-action model with open-world generalization. arXiv preprint arXiv:2504.16054,2025.
>
> >[8]. Shengliang Deng, Mi Yan, Songlin Wei, Haixin Ma, Yuxin Yang, Jiayi Chen, Zhiqi Zhang,Taoyu Yang, Xuheng Zhang, Heming Cui, et al. Graspvla: a grasping foundation model pre-trained on billion-scale synthetic action data. arXiv preprint arXiv:2505.03233, 2025.
>
> >$\large\textcolor{brown}{Weakness \ 3:}$ **The typo of "transformer".**
>
> We will revise this typo in the manuscript.
>
> >$\large\textcolor{brown}{Problem1:}$ **Why do not use SigLIP or DINOv2?**
>
> Employing a more powerful visual backbone is orthogonal to our contribution.
> Therefore, we kept the visual encoder identical to Seer’s MAE [41].
> Using the same MAE backbone eliminates confounding factors when we compare against Seer; gains can be attributed purely to our comprehensive knowledge prediction mechanism.
> We could integrate the SigLIP or DINOv2 encoder in the future.
>
>
> >$\large\textcolor{brown}{Problem2:}$ **The concern of batch size.**
>
> We apologize for the confusion: 8 is the per-GPU batch size; with 8 A100-80 GB GPUs, the global batch size is 64.

---

> ### Comment · Area_Chair_GWtm · 2025-08-05
>
> Dear Reviewer Pixv
>
> The author-reviewer discussion period will end in August 6. Please read the rebuttal, post any remaining questions or comments, and confirm the Mandatory Acknowledgement.
>
> Best regards,
> Lin

---

### Decision · Program_Chairs · 2025-09-17

**Decision:**

Accept (poster)

**Comment:**

DreamVLA propose a vision language action model for robotic manipulation that introduces additional training objectives that predict world knowledge that's relevant to robot execution. With the world knowledge prediction ability, the proposed VLA model is able to improve the genralization and reasoning abilities.

Alll the reviewers appreciate the technical novelties and the in-depth theoretical analysis. And the design, such as the dynamic regional guidance technically novel. The experiments and ablation studies have been well designed to support the proposed archetecure.

And the review comments from the reviewers’ have been clearly addressed. All the reviewers express positive attitude towards the acceptance of the paper.